# A plant-like mechanism coupling m6A reading to polyadenylation safeguards transcriptome integrity and developmental gene partitioning in *Toxoplasma*

Dayana C Farhat[1], Matthew W Bowler[2], Guillaume Communie[3], Dominique Pontier[4], Lucid Belmudes[5], Caroline Mas[6], Charlotte Corrao[1], Yohann Couté[5], Alexandre Bougdour[1], Thierry Lagrange[4], Mohamed-Ali Hakimi[1†]*, Christopher Swale[1†]*

[1]IAB,Team Host-Pathogen Interactions & Immunity to Infection, INSERMU1209, CNRSUMR5309, Grenoble Alpes University, Grenoble, France; [2]European Molecular Biology Laboratory, Grenoble, France; [3]Institut Laue-Langevin, Grenoble, France; [4]Laboratoire Génome et Développement des Plantes (LGDP), UMR5096, Centre National de la Recherche Scientifique (CNRS), Université de Perpignan via Domitia (UPVD), Perpignan, France; [5]Univ. Grenoble Alpes, INSERM, CEA, UMR BioSanté U1292, CNRS, CEA, Grenoble, France; [6]Integrated Structural Biology Grenoble (ISBG) CNRS, CEA, Université Grenoble Alpes, EMBL, Grenoble, France

*For correspondence:
mohamed-ali.hakimi@inserm.fr (M-AH);
christopher.swale@univ-grenoble-alpes.fr (CS)

†These authors contributed equally to this work

Competing interests: The authors declare that no competing interests exist.

**Abstract** Correct 3'end processing of mRNAs is one of the regulatory cornerstones of gene expression. In a parasite that must adapt to the regulatory requirements of its multi-host life style, there is a need to adopt additional means to partition the distinct transcriptional signatures of the closely and tandemly arranged stage-specific genes. In this study, we report our findings in *T. gondii* of an m6A-dependent 3'end polyadenylation serving as a transcriptional barrier at these *loci*. We identify the core polyadenylation complex within *T. gondii* and establish CPSF4 as a reader for m6A-modified mRNAs, via a YTH domain within its C-terminus, a feature which is shared with plants. We bring evidence of the specificity of this interaction both biochemically, and by determining the crystal structure at high resolution of the *T. gondii* CPSF4-YTH in complex with an m6A-modified RNA. We show that the loss of m6A, both at the level of its deposition or its recognition is associated with an increase in aberrantly elongated chimeric mRNAs emanating from impaired transcriptional termination, a phenotype previously noticed in the plant model *Arabidopsis thaliana*. Nanopore direct RNA sequencing shows the occurrence of transcriptional read-through breaching into downstream repressed stage-specific genes, in the absence of either CPSF4 or the m6A RNA methylase components in both *T. gondii* and *A. thaliana*. Taken together, our results shed light on an essential regulatory mechanism coupling the pathways of m6A metabolism directly to the cleavage and polyadenylation processes, one that interestingly seem to serve, in both *T. gondii* and *A. thaliana*, as a guardian against aberrant transcriptional read-throughs.

## Introduction

A member of the phylum *Apicomplexa*, *Toxoplasma gondii* is an obligate parasite that develops and proliferates inside a surrogate host cell and causes toxoplasmosis, a usually mild disease in

immunocompetent humans that can turn into a major threat to the unborn and to immunocompromised people for example with acquired immunodeficiency syndrome or under chemo- and graft rejection therapies (*Milne et al., 2020*). *T. gondii* has evolved dynamic and robust mechanisms for adapting and regulating its gene expression in response to the distinct external cues from the different host cell environments that the parasite faces throughout its life cycle, as well as to guide developmental transitions that are central to the parasite's persistence and transmission.

The parasite therefore faces the challenge of limiting and coordinating its transcriptional potentials in a way to swiftly adjust gene expression to its corresponding developmental requirements. However, the paucity both in numbers and variety of specific transcription factors (*Bozdech et al., 2003*), relatively to the high number of protein-encoding genes, leaves the possibility of alternative mechanisms open. Apicomplexans often display remarkably decondensed states of their chromatin, with limited heterochromatic regions as well as a general transcriptional permissiveness thus underlining the flexibility of the parasite in its dynamic life cycle requirements. A large part of the gene silencing machinery seems to be governed by the action of chromatin shapers, which were assigned significant roles in directing developmental trajectories and sexual commitment (*Farhat et al., 2020*; *Waldman et al., 2020*). Along these lines, epigenetic changes are acknowledged as driving regulators of gene expression at the level of transcription. However, the post-transcriptional level of regulation in Apicomplexans seems to be held by complex mechanisms which can be attributed to the apparent episodic absence of correlation between the levels of mRNA and their corresponding proteins at a given stage (*Holmes et al., 2017*).

While post-transcriptional control of gene expression is critical for development and environmental adaptation, the mechanisms underlying these processes are yet to be uncovered in *Apicomplexa*. In eukaryotes, capping, splicing, and 3′-end processing occur co-transcriptionally on the nascent transcripts produced by RNA polymerase II (Pol-II). Pre-mRNAs are capped at their 5′ ends and polyadenylated at their 3′ ends, and spliced, before being exported from the nucleus to the cytoplasm. This 3′ end processing is a two-step reaction in which the cleavage and polyadenylation specificity factor (CPSF) multiprotein complex catalyzes the endonucleolytic cleavage and addition of a poly(A) tail at the 3′ end of the pre-mRNA, a modification necessary for the stability and nuclear export of mature mRNAs.

Covalent modifications are also commonly found in RNAs and emerge as an alternative way of controlling the processing, stability, localization, and translatability of mRNAs. *N*6-Methyladenosine (m6A) is established as the most abundant epitranscriptomic modification of eukaryotic RNA, occurring preferentially at the conserved RRACH motif (where R = G or A; H = A, C, or U) and accumulating preferentially in 3′ UTRs (*Linder et al., 2015*; *Parker et al., 2020*; *Schwartz et al., 2013*). Nuclear-based mRNA m6A deposition is a co-transcriptional event driven by an evolutionarily conserved writer complex grouping a catalytically active m6A methyltransferase (METTL3, methyltransferase-like 3, also known as MT-A70), a second methyltransferase-like protein (METTL14) and the regulatory subunit WTAP (Wilms-tumour-1 associated protein) (*Meyer and Jaffrey, 2017*). METTL14 is reported to not hold catalytic activity itself (*Śledź and Jinek, 2016*), but to facilitate the allosteric activation of METTL3 by providing an RNA-binding scaffold (*Wang et al., 2016a*). WTAP mediates the localization of the m6A writer complex into nuclear speckles that are enriched in proteins involved in RNA processing and alternative splicing. It also recruits target RNAs, thus indirectly enhancing the catalytic capacity of the writer complex (*Ping et al., 2014*).

The disruption of the methyl enzymatic transfer in the course of forming m6A, a process that is mostly catalyzed by the activity of METTL3, has been linked to severe defects in the levels of sporulation and seed development in yeast and plants, respectively (*Clancy, 2002*; *Schwartz et al., 2013*; *Zhong et al., 2008*). Reports recently emerged linking the m6A metabolism to differentiation processes in hematopoietic (*Lee et al., 2019*) and embryonic stem cells (*Geula et al., 2015*; *Wang et al., 2014*), along with attributed roles in certain types of cancer, notably acute myeloid leukemia (*Barbieri et al., 2017*; *Vu et al., 2017*) and glioblastoma (*Cui et al., 2017*).

The recognition of the m6A modification is mostly mediated by YT521-B homology domain proteins (YTH domains) (*Meyer and Jaffrey, 2017*) where an aromatic cage is thought to recognize the modified RNA (*Luo and Tong, 2014*). Two distinct phylogenetic subfamilies of YTH domain proteins can be distinguished, exemplified by the mammalian YTHDF and YTHDC classes (*Patil et al., 2018*) which were observed to compartmentalize in the cytoplasm and nucleus, respectively, and to form dedicated sub-compartments through liquid-liquid phase separation (*Fu and Zhuang, 2020*;

*Ries et al., 2019*). These YTH readers preferentially bind methylated RNA and execute regulatory actions at the level of mRNA fate and downstream pathways. In the nucleus, YTHDC1 was shown to enhance mRNA splicing (*Xiao et al., 2016*), export (*Roundtree et al., 2017*) and degradation. In the cytoplasm, YTHDF proteins have been shown to promote mRNA translation (*Wang et al., 2015*), or conversely, mRNA decay (*Wang et al., 2014*; *Zaccara and Jaffrey, 2020*).

A unique protein arrangement bringing together a C-terminal YTH domain with N-terminal CCCH zinc finger motifs has been detected in apicomplexans, but not exclusively, as it seems to also be conserved in higher plants (*Stevens et al., 2018*). Although this co-occurrence seems to be unique to these species, the zinc finger motifs in question represent the canonical domain found in all eukaryotic counterparts of the CPSF4 (alias CPSF30; cleavage and polyadenylation specificity factor subunit 4) proteins. It is worth noting that apicomplexan and plant species have a high number of shared proteins architectures, an evolutionary consequence of an early algae-related endosymbiotic event. An archetype are the plant-like Apetala-2 (AP2) factors, which have a high regulatory potential in apicomplexans (*Jeninga et al., 2019*).

Although the role of m6A in mRNA stability and translation has been well documented, less is known regarding its impact on 3'-end processing, despite its prevalence within mRNA 3'-UTR. In animals, mutants defective in components of the m6A pathway show opposing effects on the choice of alternative polyadenylation (APA) sites (*Kasowitz et al., 2018*; *Yue et al., 2018*). It has been recently revealed in plants, that the adenosines of the consensus polyadenylation signals (PAS) motif consisting of AAUAAA, could themselves be methylated, as a nanopore-based analysis indicated an enrichment of PAS motifs around m6A motifs (*Parker et al., 2020*). A link between the presence of an m6A site and the overrun of the respective proximal PAS by the 3'end processing machinery was briefly implied in plants (*Parker et al., 2020*). Moreover, the fact that chimeric mRNAs were generated in plants, in the context of a deficiency in the CPSF30-YTH (CPSF30L) isoform, hints at transcriptional readthrough events taking place, and at the involvement of the YTH domain of CPSF30 in this process (*Pontier et al., 2019*). While a link between m6A-related proteins and 3'end processing players has been proposed, the mechanistic and functional outcomes of such a cross between these two pathways as well as their evolution across species have not yet been fully explored.

Here, we describe the *T. gondii* homolog of CPSF4 and we demonstrate by mass spectrometry, the involvement of the CPSF4-YTH protein in the core CPSF complex, as well as providing the overall composition of the latter through the use of endogenously tagged and purified putative CPSF subunits. More importantly, in vitro evidence shows the ability of the *T. gondii* YTH domain to recognize m6A-modified RNAs, which we corroborate by providing comparable data in *Arabidopsis thaliana*. We were also able to determine the crystallographic structure of the *T. gondii* YTH domain in complex with a short seven mer m6A-modified RNA. Finally, our native RNA sequencing analysis allowed us to characterize for the first time the m6A modification landscape of *T. gondii*, and second to shed light on an essential regulatory mechanism coupling the pathways of m6A metabolism with the polyadenylation processes, one that interestingly seem to serve, in both *T. gondii* and *A. thaliana*, as a guardian against aberrant transcriptional read-throughs.

## Results

### Architecture of the *T. gondii* pre-mRNA 3' processing complex

Our discovery of CPSF3 (also known as CPSF73) as a promising therapeutic target against life threatening apicomplexan parasites (*Bellini et al., 2020*; *Palencia et al., 2017*; *Swale et al., 2019*), prompted us to pursue a better understanding of 3'processing of mRNAs in *Apicomplexa*. The overall architecture of the CPSF complex in *Apicomplexa* is still debated; unlike animals, plants, and yeast, for which the polyadenylating complexes have been well characterized. Apart from bioinformatics-based identification of some of the subunits (*Ospina-Villa et al., 2020*; *Stevens et al., 2018*), no direct biochemical evidence has yet been established to support the interaction of these proteins in-vivo. In order to define the subunit composition of the CPSF complex in *T. gondii*, we proceeded by endogenously tagging (C-terminal HA-FLAG) and probing several putative subunits of the latter. These included CPSF1, CPSF3, CPSF2, CPSF4 also known and recognized as CPSF-160, CPSF-73, CPSF-100, CPSF-30, respectively. These latter proteins along with Symplekin, Fip1, WDR33 and the

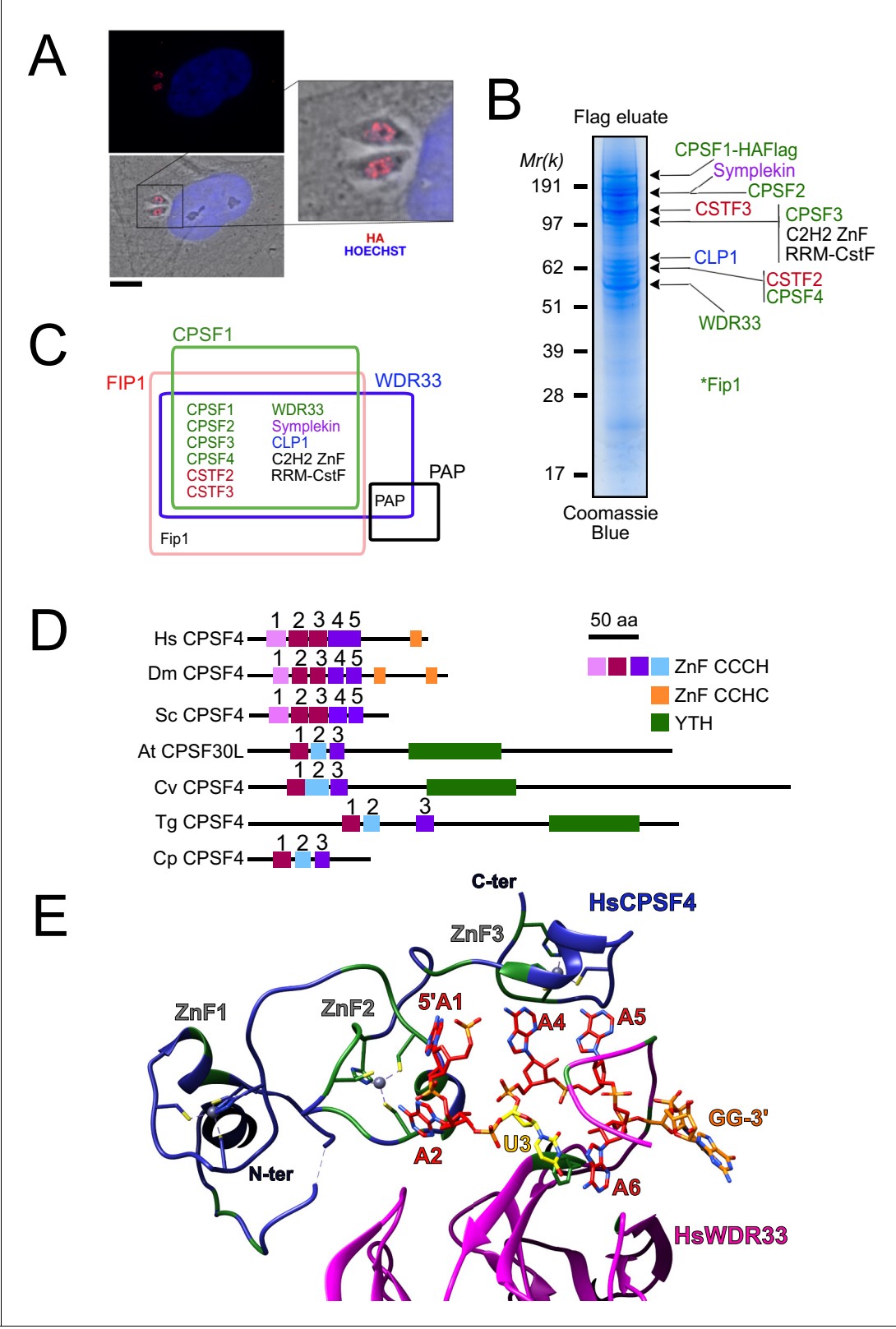

**Figure 1.** The nuclear-based CPSF subunit TgCPSF4 harbors a cross-phyla functional domain conservation, except for its additional plant-like YTH domain. (**A**) A representative image of the nuclear staining CPSF1 (red) in human primary fibroblasts (HFFs) infected with parasites expressing an HA–Flag-tagged copy of CPSF1. Cells were co-stained with Hoechst DNA-specific dye. Scale bar, 10 μm. (**B**) MS-based proteomic analysis of the CPSF1-Flag elution identified many of the CPSF complex subunits. The identities of the proteins are indicated on the right. Fip1 was detected in sub-

*Figure 1 continued on next page*

*Figure 1 continued*

stoichiometric quantities in higher molecular weight band extractions but no band at its predicted size was cut-out for analysis. (C) Venn diagram showing the overlap of the proteins identified by mass spectrometry in the CPSF1, WDR33, FIP1, and PAP pulldowns. (D) Domain architectures representation of CPSF4 homologues. Hs *Homo sapiens*, Dm *Drosophila melanogaster*, Sc *Saccharomyces cerevisiae*, At *Arabidopsis thaliana*, Cv *Chromera velia*, Tg *Toxoplasma gondii*, Cp *Cryptosporidium parvum*. CPSF4 in *T. gondii* is encoded by *TGME49_201200*. (E) An adapted representation of the HsCPSF4 recognition of the polyadenylation signals (PAS) consisting of the hexamer motif AAUAAA, showing the ZNF2 binding to A1 and A2. HsWDR33 and HsCPSF4 are shown in cartoon fashion in magenta and blue respectively. The AAUAAA RNA is shown in stick fashion. Strict sequence conservation of RNA binding residues with *Toxoplasma gondii* homologs is shown in green.

The online version of this article includes the following figure supplement(s) for figure 1:

**Figure supplement 1.** The CPSF subunits are similarly nuclear-based, essential for parasite's fitness, and expressed mostly outside of the latent parasitic stages.

putative poly (A) polymerase (PAP) partner which we also probed, all displayed nuclear staining by immunofluorescence (*Figure 1A*, *Figure 1—figure supplement 1A*).

Among these, CPSF1, WDR33, and Fip1 provided the clearest FLAG-mediated immunoprecipitation data (*Figure 1B–C*, *Supplementary file 1*), with CPSF1 displaying the most discernible and intact complex, when analyzed by band-specific mass spectrometry-based proteomics (*Figure 1B–C*, *Supplementary file 1*), which demonstrated the highest levels of abundance of the putative candidates within the bands corresponding to their respective predicted molecular weights. For instance, the CPSF4 subunit with its theoretical mass of 68 kDa, can be found most abundant within the band at 62 kDa. A relatively high quantity was detected of two as yet unknown proteins which may constitute apicomplexan-specific subunits of the CSTF or CPSF complexes, namely TGME49_261960 carrying an RNA recognition motif, and TGME49_254210 carrying a C2H2 zing finger domain. Similarly, the CPSF complex core components were also pulled down during the purification of the Fip1 subunit, however less rigorously as can be observed from the mass spectrometry-based proteomic characterization (*Figure 1B–C* and *Supplementary file 1*). It is worth noting that the mostly non-structured nature of the Fip1 protein could suggest some degradation events taking place to justify the relatively poor peptide representativity of this subunit in the CPSF1 immunoprecipitation profiling (*Figure 1C* and *Supplementary file 1*). None of these subunits immunoprecipitation data allowed the detection of the PAP protein, despite the fact that we managed to purify the PAP-FLAG protein separately, except for WDR33 interactome analysis in which only very weak amounts of PAP were detected (*Figure 1B–C* and *Supplementary file 1*). This could be explained either by a highly transient binding mode of PAP, or by its featuring of weaker interactions within the complex that could have been disrupted during the stringent salt washing conditions (up to 500 mM KCl) of the purification steps.

Other than all the identified CPSF subunits seemingly sharing a nuclear-based localization, their tachyzoite-based fitness assessment suggests that they are all essential for the survival of the parasite (*Figure 1—figure supplement 1B*). This goes in accordance with the consistent level of expression of all the subunits, in the tachyzoite stage and other actively proliferating stages, while plotting their average expression profiles revealed markedly levels in latent stages that is cyst and immature oocyst stages (*Figure 1—figure supplement 1C*).

## *T. gondii* CPSF4 harbors a YTH domain in addition to the conserved zinc fingers, an architecture also found in plants

In the CPSF complex, the *T. gondii* CPSF4 subunit can be distinguished as one holding a unique architecture which interestingly is shared with the plant CPSF30L family, and it constitutes a co-occurrence of three zinc fingers and a conserved YTH domain (*Figure 1D*). In comparison, the metazoan and fungi counterparts display five distinctive and evolutionary-conserved CCCH-type zinc fingers, in addition to a zinc knuckle, but none of them present the YTH domain within the same protein (*Figure 1D*). This architecture is detected on one of two isoforms of the CPSF homolog (CPSF30 gene *At1g30460*) in *Arabidopsis thaliana*, namely the CPSF30L, while the short version CPSF30S is one that lacks the YTH domain (*Figure 1—figure supplement 1D*; *Chakrabarti and Hunt, 2015*; *Delaney et al., 2006*; *Liu et al., 2014*). The alternative splicing and polyadenylation events that form these double plant isoforms, is not seen in the *T. gondii* homolog which is

expressed in a constitutive manner throughout the parasite life cycle (*Figure 1—figure supplement 1C*), and is consistently predicted as a 62 kDa protein.

Evolutionarily, some degree of conservation between the different zinc fingers of the CPSF4/CPSF-30 homologues can be recognized (*Figure 1D–E* and *Figure 1—figure supplement 1E*). For the following assessment, the proteins of metazoans and fungi, can be set together against those of plants, *Apicomplexa*, but also of chromerids which constitutes one of the last common ancestors between the two. These proteins can be compared in view of the ability of the human CPSF4 zinc finger motifs to recognize the canonical polyadenylation signals (PAS) consisting of the hexamer motif AAUAAA, with which binding is sufficient to recruit poly(A) polymerase through CPSF30 (*Figure 1E*; *Clerici et al., 2018*; *Sun et al., 2018*). A close-up view of the CCCH-type zinc fingers highlights conservation between the ZNF2 from metazoa and fungi and the ZNF1 from plants, chromerids and *Apicomplexa* (*Figure 1—figure supplement 1E*), suggesting that the function acknowledged for the metazoan CPSF4 might be conserved in the aforementioned counterparts, that former being an ability to recognize nucleotides A1 and A2 (*Figure 1E*). Similarly, the ZNF3 from plants, chromerids and *Apicomplexa* is conserved with the ZNF5 from metazoa and fungi, a motif involved in Fip1 recruitment (*Hamilton et al., 2019*).

## A nuclear-based m6A catalytic core complex in *T. gondii* encompassing both conventional and specific subunits

The presence of a YTH domain within *T. gondii* CPSF4 was intriguing and prompted us to explore its link to m6A, as it is recognized as a reader of this modification. First, we checked for the corresponding methyltransferases (writers) in *T. gondii*. As with many *Apicomplexa,* the *T. gondii* genome has retained the genes encoding for METTL3 and METTL14 which together are known to form a core catalytic complex, noting also the conservation of the regulatory subunit WTAP (*Figure 2* and *Figure 2—figure supplement 1A*; *Baumgarten et al., 2019*). Sequence analysis suggests that *T. gondii* METTL3 has an active catalytic site while TgMETTL14 displays a disrupted SAM-binding motif, suggesting catalytic inactivity shown in *Wang et al., 2016a*, which is in agreement with the current models of METTL14 serving as an RNA-binding platform activating allosterically the catalytically active METTL3 (*Wang et al., 2016a*; *Wang et al., 2016b*). Using bioinformatic analysis, we failed to detect in apicomplexan genomes the auxiliary proteins that are usually found in the identified human complexes, and that are thought to aid the catalytic core components in the correct m6A deposition (*Figure 2—figure supplement 1A*; *Balacco and Soller, 2019*).

To further explore how the enzymes partner in vivo, we generated knock-in parasite lines expressing a tagged version of METTL3, METTL14, and WTAP. Immunofluorescence analysis of intracellular parasites revealed an almost exclusive nuclear staining for all of METTL3, METTL14, and WTAP (*Figure 2A–B*). Intense punctate foci were detected, similarly to their human counterparts which were seen to accumulate as condensates within nuclear speckles (*Ping et al., 2014*), with these latter representing phase-separated membrane-less organelles enriched in pre-mRNA splicing factors. In addition to its nuclear staining, the WTAP protein displayed a diffused staining throughout the cytoplasm, hinting at the ability of this protein to shuttle between the nucleus and the cytoplasm (*Figure 2A*).

In order to validate the predicted association between METTL3, METTL14, and WTAP, and in the hope of identifying auxiliary proteins, even if divergent ones, we opted for a biochemical approach, which allowed us to define the interactome of each of the catalytic core subunits, using the respective endogenously HA-FLAG-tagged knock-in parasites. Western blotting of the FLAG eluates revealed a single band at the expected size for each protein, with the exception of METTL3 which exhibited lower substoichiometric forms, which may result from a sensitivity to degradation (*Figure 2—figure supplement 1B*). Coomassie stain analysis of the FLAG eluates suggested that all three proteins bind to multiple partners under high stringent wash conditions (0.5 M NaCl and 0.1% NP-40; *Figure 2C*). These partnerships were subsequently resolved by mass spectrometry-based proteomics which identified METTL3 and METTL14 as an intact RNA methyltransferase core complex (*Figure 2C*) with an apparent molecular weight of 400–500 kDa by size exclusion chromatography (*Figure 2D*). Although METTL14 was not detected in the eluates of WTAP and vice versa, WTAP was found in the METTL3 pull-down in significant quantities despite the stringent washing conditions (*Supplementary file 1*). Additional partners were recognized as the RNA-binding proteins displaying multiple RRM (TGME49_291930) or KH (TGME49_235930) domains, an ATP-dependent RNA

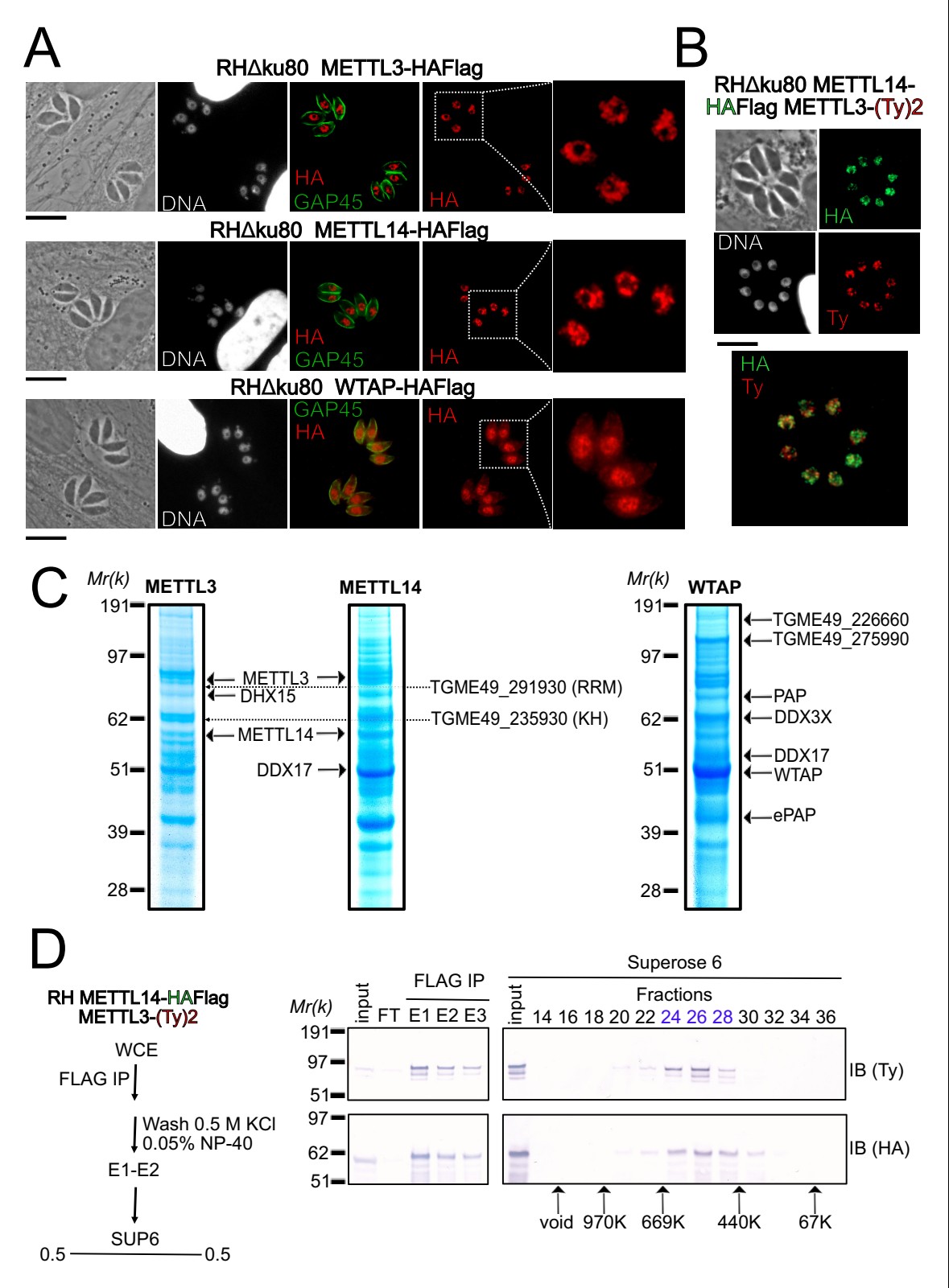

**Figure 2.** A nuclear-based m6A catalyzing complex in *T. gondii* incorporates both conventional and specific subunits. (**A**) IFA showing the nuclear localization of METTL3, METTL14 and WTAP (red), using human primary fibroblasts (HFFs) infected with the corresponding parasites harboring endogenously HA-FLAG-tagged proteins. The parasitic membrane is probed using anti-GAP45 antibodies (green). Cells were co-stained with Hoechst DNA-specific dye. Scale bar, 10 μm. (**B**) IFA of HFFs that were infected with parasites harboring METTL3 endogenously tagged with Ty within the RH

*Figure 2 continued on next page*

Figure 2 continued

METTL14–HAFlag lineage. Fixed and permeabilized parasites were probed with antibodies against HA (green) and Ty (red). Scale bar, 10 μm. (C) Coomassie Blue staining of the eluates used for subsequent MS-based proteomic analysis for the identification of the interactomes of METTL3, METTL14, and WTAP. The identities of the proteins are indicated. (D) METTL14 was FLAG-affinity purified from whole cell extract of parasites co-expressing METTL14-HAFlag- and METTL3-(Ty)2-expressing parasite with Flag affinity. Flag-eluted peptides were fractionated on a Superose 6 gel filtration column in the presence of 0.5 M KCl. Flag chromatography and gel filtration fractions were separated through SDS-polyacrylamide gel and analyzed by western blot with anti-HA and anti-Ty antibodies. Fraction numbers are indicated on top of the gel.

The online version of this article includes the following source data and figure supplement(s) for figure 2:

**Source data 1.** Uncropped immunoblots: uncropped western blots corresponding to *Figure 2D*.

**Figure supplement 1.** A set of conserved and fitness conferring m6A-related enzymes can be detected in the *Apicomplexa* phylum.

**Figure supplement 1—source data 1.** Uncropped immunoblots: uncropped western blots corresponding to *Figure 2—figure supplement 1B*.

helicase involved in pre-mRNA splicing (DHX15 and DDX17) and the notable PAP enzyme (*Figure 2C*). Interestingly, our experiments also reveal the existence of two new partners, the uncharacterized proteins TGME49_226660 and TGME49_275990, which were detected abundantly in WTAP eluates but also in less quantity in METTL3 and METTL14 pull-down (*Figure 2C*).

## A sustained depletion of METTL3 drastically impairs intracellular m6A distribution

Having identified the complexes putatively acting as methyltransferases of the m6A modification, we next examined the extent of this supposition by attempting to deplete METTL3 function as it is thought to carry the catalytic potential of the core complex. To this end, we employed the auxin-inducible degron (AID) system, for an acute and reversible depletion of METTL3, owing to its requirement for the fitness of the tachyzoite (*Figure 2—figure supplement 1C*). Treatment of the edited parasites with indole-3-acetic acid (IAA) triggered a highly specific and near-complete clearance of the pool of METTL3–mAID–HA protein after 24 hr (*Figure 3A–B*). The m6A mark is detected mostly within the cytoplasm by immunofluorescence staining in tachyzoites (*Figure 3C*). Consistent with its role as a catalytically active subunit, METTL3 knockdown triggers a significant reduction in m6A levels as early as 12 hr after IAA treatment, with a more dramatic decrease after 24 hr (*Figure 3C–D*). It should be noted that along with this post-translational loss of METTL3, a CRISPR-based transitory genetic inactivation of this protein resulted in a similarly significant drop in the m6A levels (*Figure 3E*).

Despite the fact that in human cells, the total m6A content in the mRNAs also fell by ~75% only after 96 hr following the triple knockdown of METTL3, METTL14, and WTAP *Ke et al., 2015*; our observations are still indicative of a high level of stability of the m6A modification which is probably maintained throughout the life of an mRNA transcript. This, in addition to the fact that as opposed to higher eukaryotes, a lack of m6A demethylases is recognized in the phylum (*Figure 2—figure supplement 1A*; *Baumgarten et al., 2019*) so that the dynamic changes of this mark would potentially be intimately linked to the activity of its corresponding writers, but also that of its YTH-domain-containing readers.

## The YTHDC-1 orthologue domain of *T. gondii* CPSF4 binds exclusively to m6A-modified RNA in vitro

YTH-containing readers play a crucial role in the recognition of m6A-modified RNA. To validate the biochemical function of the YTH domain contained within CPSF4 (predicted from residues 434 to 598), we undertook to recombinantly express the domain in *E. coli* with an N-terminal TEV cleavable 8*His tag with minimized extremities so as to limit disordered regions and obtained pure and mono-disperse preparation of the protein (*Figure 4A*). We then used isothermal calorimetry to titrate a chemically synthetized seven mer RNA with a consensus m6A site (5'-GAACAUU-3') possessing or lacking the m6A modification (*Figure 4B*). As measured, binding towards the RNA substrate has a relatively high dissociation constant (Kd) but is entirely dependent on the presence of an m6A modification as almost no binding affinity is measured in the un-modified RNA (*Figure 4B*). The same is also true for the YTH module (residues 277–445) of *Arabidopsis thaliana* CPSF30 (*Figure 4C*)

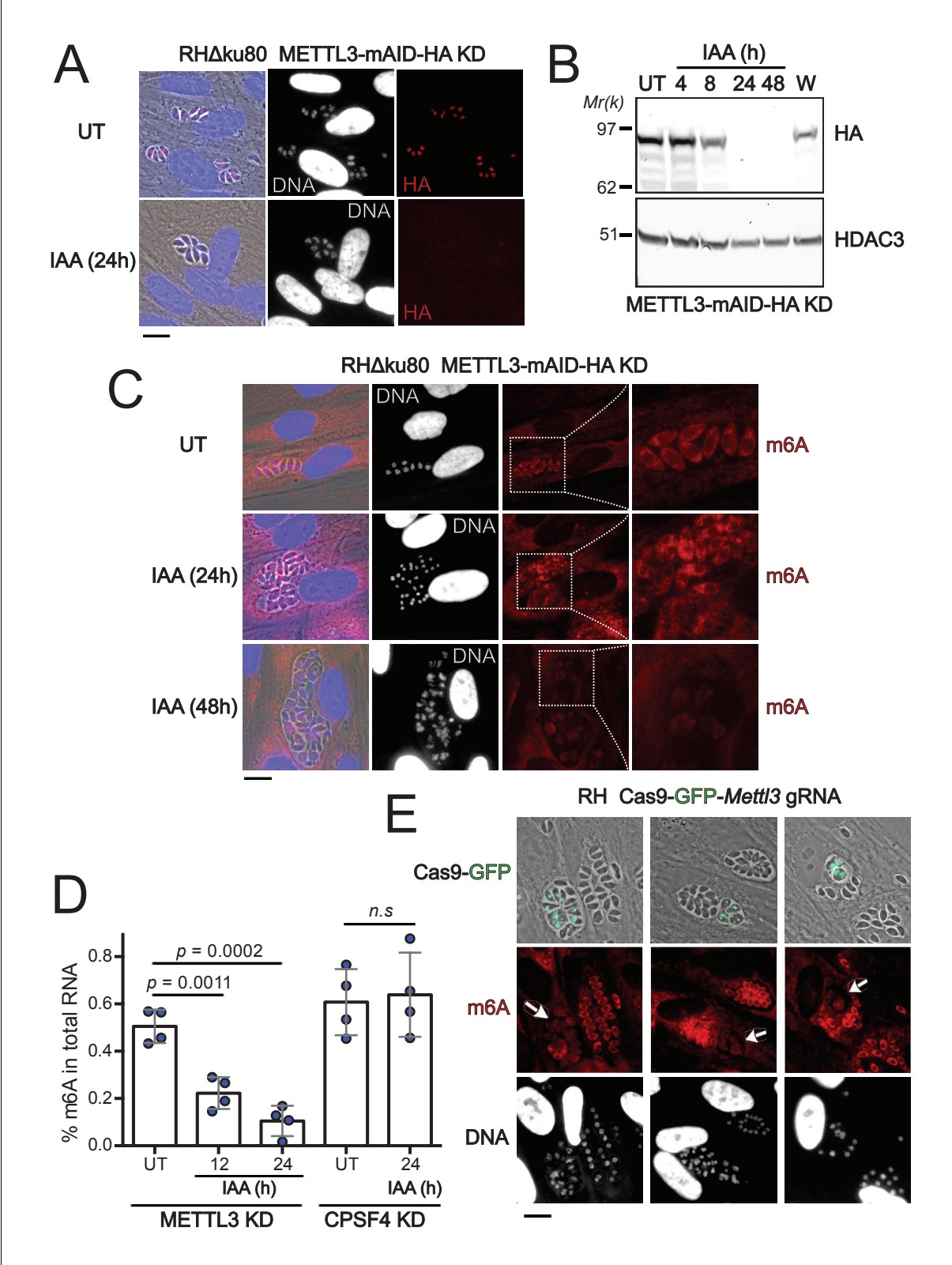

**Figure 3.** The depletion of METTL3, both post-translationally or genetically, impairs the level of m6A. (**A**) METTL3 protein expression levels after 24 hr of adding IAA, displayed by IFA on HFF cells infected with RH parasites engineered to allow the degradation of the endogenously tagged METTL3-mAID-HA. Cells were probed with antibodies against HA (red) and DNA was stained using the Hoechst DNA dye. Scale bar, 10 µm. (**B**) Time-course analysis of the expression levels of METTL3–mAID–HA. The samples were taken at the indicated time periods after addition of IAA and were probed

*Figure 3 continued*

with antibodies against HA and HDAC3. IAA-treated (24 hr) parasites were also washed (**W**), incubated with fresh media in the absence of IAA (12 hr) and analyzed using western blot. The same experiment was repeated two times and a representative blot is shown. (**C**) The effect of METTL3 depletion on the m6A levels, detected upon 24 and 48 hr of IAA-dependent Knock-Down induction. Specific antibodies were used to probe the m6A mark. The DNA staining points at defects at the nuclear level following the METTL3 depletion. (**D**) Quantitation of m6A levels by ELISA in total RNA from METTL3–mAID–HA and CPSF30-mAID-HA purified tachyzoites untreated (UT) or treated with IAA at the indicated time periods. Data are the mean ± s.d. of three biological replicates. p Values were calculated using two-tailed unpaired Student's t-test. (**E**) g-RNA targeted against the *METTL3* gene allows the genetic inactivation of this latter, allowing to detect the effects of this disruption on the levels of m6A (in red) within the parasites that were touched by the Cas9 (marked with arrows). The efficiency of genetic disruption in Cas9-expressing parasites was monitored by cas9-GFP expression (in green). Scale bar, 10 μm.

The online version of this article includes the following source data for figure 3:

**Source data 1.** Uncropped immunoblots: uncropped western blots corresponding to *Figure 3B*.

confirming that this ability to bind m6A is a shared evolutionary feature across the Apicomplexa and plant kingdom.

## Crystal structure of the CPSF4 YTH module reveals a highly conserved binding pocket for m6A

To further validate the function of the YTH domain of CPSF4, we undertook structural characterization of this domain using X-ray crystallography. TgCPSF4-YTH (aa 434–598), as used in the ITC experiments generated distinct crystal forms: Apo, m6A bound and m6A modified 7-mer RNA bound (*Supplementary file 2*) which all diffracted to high resolutions (up to 1.45 Å for the RNA bound form), sometimes using fully automated upstream crystal harvesting (using the EMBL crystal direct technology) and automated crystal diffraction (using MASSIF-1 at the ESRF). Molecular replacement (using the YTHDC1 pdb **4r3i**) was able in all cases to rapidly find phasing solutions. Overall, the CPSF4 YTH domain folds into a well-structured domain (*Figure 5A*) featuring six alpha helices (α1–6) and five beta sheets (β1–5). The m6A-binding site involves residues in, or close to, helices α1/ α2/ α3 and beta sheet 1 (*Figure 5A* and *Figure 5—figure supplement 1A*). For convenience, we depict the m6-adenosine as an adenine, although in practice crystals were co-grown with m6-adenosine, the ribose electron density is indeed poorly visible in the crystal structure, the adenine electron density is however unequivocal (*Figure 5—figure supplement 1B*). The binding of m6A favors crystal growth in *P1* symmetry instead of a *P4₁2₁2* symmetry, however, the binding event does not induce important conformational changes within the protein (*Figure 5—figure supplement 1C*), the N-terminal residues (ranging from 437 to 446) fold as random coils on opposite sides, most likely as a consequence of crystal packing.

When comparing the CPSF4 YTH domain with its closest homologue structure (human YTHDC1), the overall general conservation of the domain is observed (*Figure 5B*) with both sharing a sequence identity of 38% and having most of their secondary structure features conserved (*Figure 5C*), apart from α3 and β1 which are unique secondary structures to CPSF4 YTH. Although strongly conserved, the aromatic cage recognizing the m6A displays a notable difference in the region between residues 519 and 526 (res 428–439 in the human YTHDC1) with the absence of a methionine residue (M434 in YTHDC1) and the presence of an additional valine (V522). Finally, visible angular differences in the planes of the m6A base between human YTHDC or CPSF4 YTH are seen in the m6A co-crystals but do not reflect a biological reality as the m6A-modified RNA with CPSF4 YTH adopts a comparable plane as that of YTHDC1.

## The m6A RNA / CPSF4 YTH structure reveals a conserved RNA-binding mode and no sequence specificity outside of m6A recognition

With no prior information on the potential sequence specificity of CPSF4 YTH toward m6A-modified RNA in *T. gondii*, we undertook to crystallize the Tg-YTH with the canonical m6A modified short RNA used in isothermal titration experiments. Although using a seven mer GA-m6A-CAUU RNA, we can only visualize the electron density of the m6A followed by two nucleotides downstream (*Figure 6A*). The RNA is bound within a clearly positively charged groove which is then followed by a potential secondary groove. In this structure, as in others for YTH domains, the m6A-modified base is twisted inwards compared to the other bases. Although the m6A base electron density is

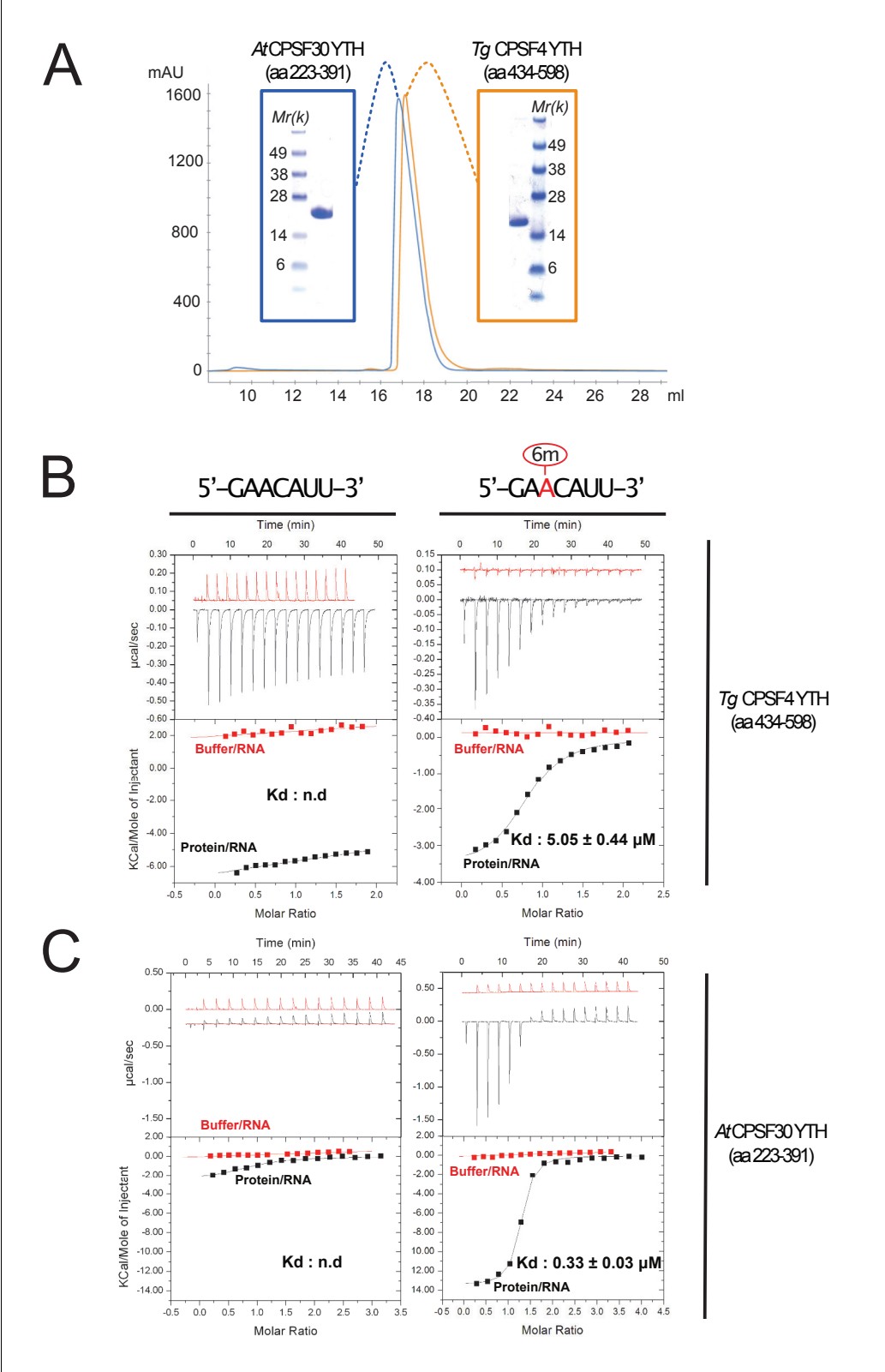

**Figure 4.** Binding of m6A-modified RNA to recombinant forms of *T. gondii* and *A. thaliana* CPSF4-YTH domains. (**A**) Final purification step gel-filtration chromatograms (using a S200 column) and associated NUPAGE gels of *T. gondii* and *A. thaliana* CPSF4-YTH domains shown in blue and orange, respectively. (**B and C**) IsoThermal Calorimetry (ITC) titrations obtained from recombinant TgCPSF4YTH (**B**) or AtCPSF4YTH (**C**) against unmodified (left panel) or m6A modified seven mer RNA (right panel). For both conditions, RNA within buffer (red curves) and RNA within protein solutions (black

*Figure 4 continued on next page*

Figure 4 continued

curves) titrations were included. Data is displayed above as µcal/sec peaks at every ligand injection as a function of Molar Ratio while integrated peak values fitted with association curves are shown below.

clearly visible, the following cytosine and adenosine have poor electron density for the bases which are solvent exposed, the sugar and phosphate backbone is however clearly visible. This feature can be explained by the relatively poor number of interactions visible between the RNA and CPSF4 YTH (*Figure 6B*) which mostly concentrate on the m6A through the hydrophobic interactions and to a lesser extent through polar interactions with the RNA phosphate and sugar backbone, notably interactions with 2' OH moieties which implies an ability to discriminate between single stranded DNA. No sequence specificity elements are visible in our structure suggesting that any sequence upstream or downstream from the m6A would bind similarly.

When comparing our model to the recently published *Arabidopsis thaliana* CPSF4-YTH domain bound to a 10 mer RNA (pdb id: 5ZUU) (*Hou et al., 2021*), which forces a dimerization of the YTH/RNA complexes through the 5'-$C^6U^7A^8G^9$-3' palindromic sequence, the backbone conformation of the downstream nucleotides after the m6A remain remarkably conserved (*Figure 6—figure supplement 1A*). This can also be observed in YTHDC1 complexes with single strand DNA (pdbid:6WEA; *Woodcock et al., 2020*) which has visible 5' nucleotides before the m6A and is coherent with the overall structural conservation of these domains (*Figure 6—figure supplement 1B*) which all display a similar positively charged binding groove dedicated to 3' binding after the m6A (*Figure 6—figure supplement 1C*). Interestingly, the *T. gondii* CPSF4-YTH domain described here is the only one to clearly show a secondary positively charged binding groove which could indicate alternative binding modes which would be unique to this YTH domain.

## The depletion of CPSF4 impairs transcription termination, as detected by Nanopore DRS

After having established the ability of the *T. gondii* CPSF4 YTH domain to specifically bind m6A-modified RNA biochemically, we wanted to tackle the functional outcome of the unique architecture of a m6A reader within this polyadenylation central subunit. In order to answer this question, we proceeded by first exploring the transcriptional outcome of depleting the CPSF4 protein by employing the auxin-inducible degron system (*Farhat et al., 2020*). After addition of IAA, the nuclear pool of CPSF4-mAID-HA slightly decreased at 8 hr but the complete clearance was not evident until 24 hr (*Figure 7A–B*). This same tool was now used to conduct a detailed time-course measurements of mRNA levels using Illumina RNA-sequencing (RNAseq) (GSE168155), which allowed the detection of a sizeable fraction of mRNAs that accumulated following the IAA-dependent depletion of CPSF4, and in a gradual manner as shown by hierarchical clustering analyses (*Figure 7C*, *Figure 7—figure supplement 1A–B* and *Supplementary file 3*).

Although some genes had their expression induced as early as after 7 hr after the IAA-dependent depletion of CPSF4 (e.g. *TGME49_208730* in *Figure 7D*), the main transcriptional phenotype pointed to possible alterations of the transcription termination occurring in the context of the KD of CPSF4. At many *loci*, the depletion of CPSF4 was accompanied by an apparent transcriptional readthrough that went beyond the annotated 3'end sequence of a certain gene to breach into the adjacent one, and that generated very long transcripts (top of *Figure 7E* and *Figure 7—figure supplement 2*). Could this defect in the transcription termination be caused by an overrun of the gene's proximal polyadenylation signal (PAS)? Nanopore long-read Direct RNA Sequencing (DRS) data argue in favor of this suggestion, and point to an alternative 3'end processing that is occurring using the downstream PAS of the adjacent gene (bottom of *Figure 7E* and *Figure 7—figure supplement 2*).

These claims are based on the fact that nanopore technology allows the direct sequencing of individual native mRNAs, as only polyadenylated RNAs can pass through the pore complex due to the ligation of motor proteins to poly(T) adaptors. Furthermore, the sequencing is stranded going from 3' to 5', so 3' ends are sequenced first. Therefore, the fact that following the depletion of CPSF4, the same aberrant transcription termination seen with *illumina* RNAseq is detected by nanopore DRS on single full length transcripts, provides evidence that the initial 3' end polyadenylation site

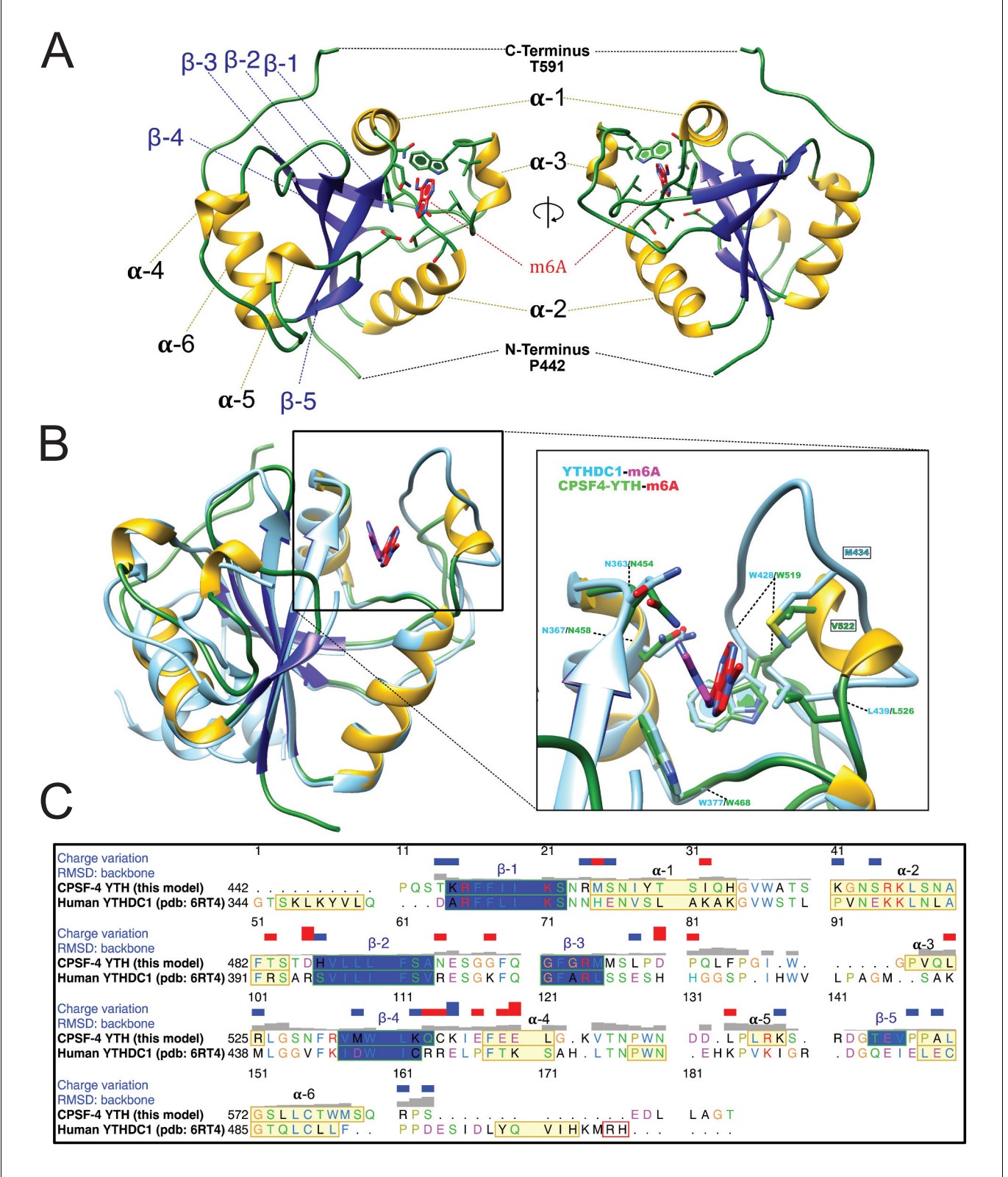

**Figure 5.** Overall structure of the CPSF4 YTH domain. (**A**) General structure architecture. The structure is displayed in a ribbon diagram with only side chains within the m6A-binding site shown. The right representation corresponds to the left representation 180° centrally rotated on itself. Alpha helices are shown in yellow, β-sheets in blue and loops and coils in green. m6A from this model in both A and B panels is shown as red sticks. (**B**) CPSF4/YTHDC1 structural comparison. Both CPSF4 YTH (colored as in panel A) and YTHDC1 (pdb 6RT4 colored in light blue) structures were chain

*Figure 5 continued on next page*

*Figure 5 continued*

superposed on their Cα backbones. m6A ligands are shown in stick representation, the one binding to YTHDC1 is colored in magenta. A blow-up panel on the left focuses on the m6A-binding site. (C) Sequence alignment depicting alpha helices in yellow, β-strands in blue displaying RMSD (backbone) per residue as well as the charge variation per residue (blue being positive and red negative). Representations, structure matching and alignments were made using UCSF Chimera.

The online version of this article includes the following figure supplement(s) for figure 5:

**Figure supplement 1.** m6A-binding site.

has been overrun, and that this process has now shifted to employ an alternative PAS within the downstream gene (bottom of *Figure 7E*), thus the display of these elongated transcripts with aberrant 3'UTRs (*Figure 7E*, *Figure 7—figure supplement 2* and *Figure 7—figure supplement 3*). Such transcriptional readthrough events were commonly detected and with great accuracy, and they suggest that the depletion of CPSF4 is leading to an impairment of a functional termination at 3' boundaries of a set of genes.

Beside the CPSF4-KD phenotype assessment, the nanopore data proved useful in identifying new isoforms of alternatively spliced transcripts in *T. gondii*, an event that seems to occur frequently, thus contributing to a higher level of proteome complexity of this parasite (*Figure 7F* and *Figure 7—figure supplement 4*). The FLAIR (Full-Length Alternative Isoform analysis of RNA; *Tang et al., 2020*) analysis, which is designed to detect, correct, and collapse splicing isoforms, provided evidence of widespread alternative splicing in *T. gondii* occurring by means of the various established AS archetypes (exon skipping, intron retention, mutually exclusive exons, alternative 5' or 3' splice sites selection, alternative transcription start sites and 3' termination ends) (*Keren et al., 2010*). Despite their broad occurrence, these AS transcripts can, in cases of most genes, be considered as a minor fraction, for most genes, in comparison to the canonical transcripts, as they displayed a lower read coverage (*Figure 7F* and *Figure 7—figure supplement 4*).

## The depletion of the m6A writer, METTL3, phenocopies that of CPSF4 in its generation of transcriptional readthrough chimeric RNAs

When alternative splicing occurs at *loci* displaying a transcriptional readthrough, the resulting alternatively spliced elongated transcripts are considered as chimeric RNAs (*Gingeras, 2009*; *Grosso et al., 2015*). We have detected chimeric RNAs at several *loci*, following the depletion of the polyadenylation subunit CPSF4 (*Figure 8* and *Figure 8—figure supplement 1*). In view of the unique architecture of this protein, combining the conserved 3'end processing zinc fingers, with the YTH that we structurally demonstrated as an m6A reader, it seemed only logical to tackle the weight of this RNA modification on the termination defects that we observed in the context of the CPSF4 KD. For this purpose, we proceeded by assessing the outcomes of diminishing this mark at the level of its deposition by employing the previously described METTL3 KD cell line (*Figure 3A–B*) to generate nanopore-sequenced RNA data, at 24 hr post induction of the knock-down, a time that is short enough to be able to discriminate primary from secondary transcriptomic effects (SRA data PRJNA705300).

Nanopore DRS analysis revealed recurrent events of transcription termination defects in METTL3-depleted cells when compared to untreated cells, at loci that exhibited similar patterns in the context of CPSF4 being depleted (*Figure 7E*, *Figure 8* and *Figure 7—figure supplement 3*). We detected and assessed the average distribution of readthrough transcripts at a genome-wide level, by using ChimerID scripts (*Parker et al., 2020*), and concluded that the formation of RNA chimeras following the depletion of CPSF4 and METTL3 occurred globally over all chromosomes (*Figure 8A*). Overall, most of the observed readthrough event within the METTL3 KD are also found in the CPSF4 KD (*Figure 8B* and *Figure 8—figure supplement 1A*). CPSF4 also has a higher number of readthrough events because of its central role to recruit the CPSF complex which can be independent of m6A (*Figure 8C* and *Figure 8—figure supplement 1B*). Importantly, this approach only quantifies readthrough events which are subsequently poly-adenylated and generally originate from strongly transcribed regions (*Figure 8D*, *Figure 8—figure supplement 1C* and *Supplementary file 4*). It should be noted that these defects were not representative cases of the frequently reported premature termination (*Kamieniarz-Gdula and Proudfoot, 2019*), but instead events of readthrough and

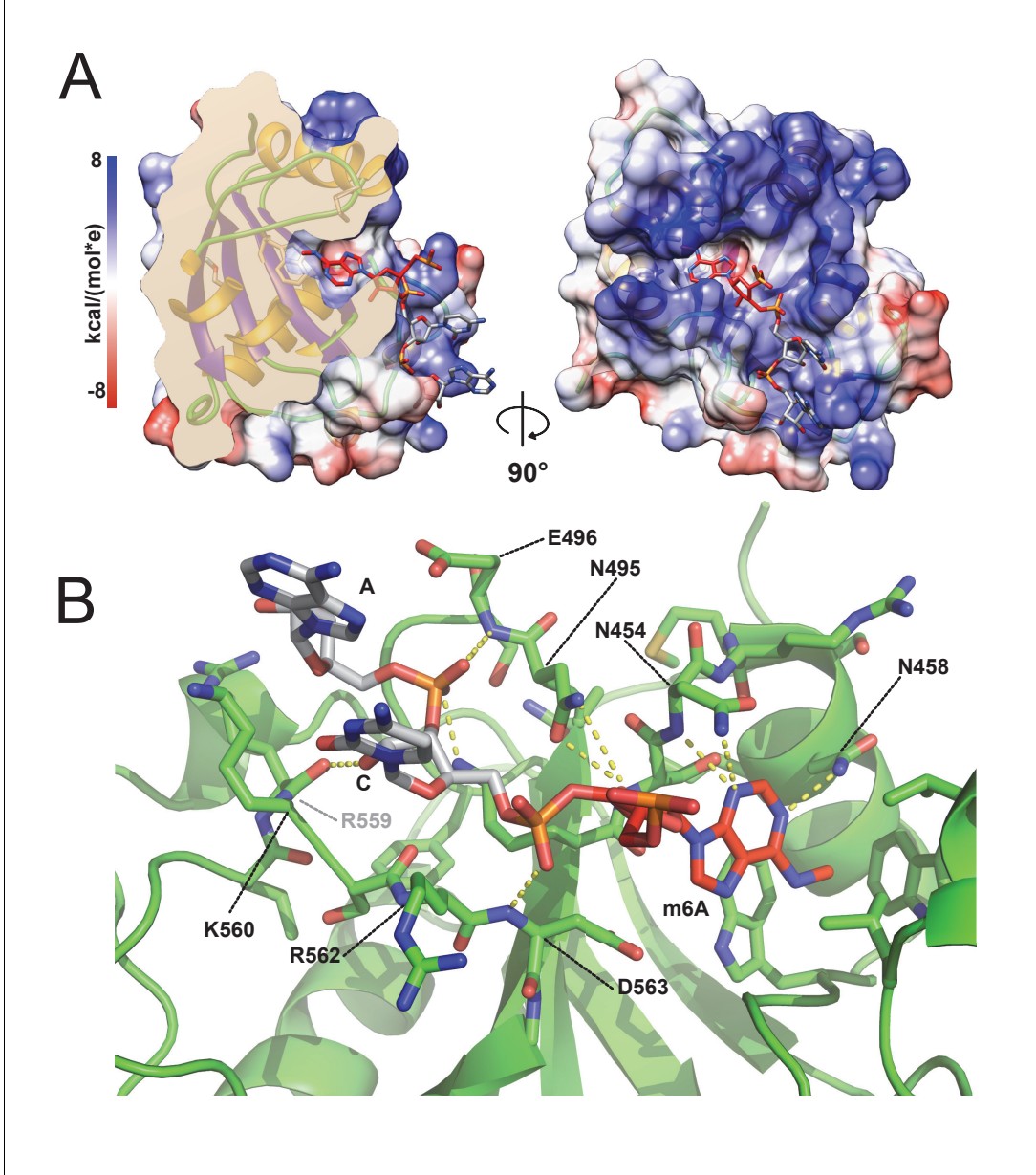

**Figure 6.** m6A RNA/CPSF-4 YTH co-crystal structure. (**A**) Semi-transparent surface representation of CPSF-4 YTH displayed with a coulumbic surface coloring (UCSF chimera), RNA bases are shown in stick representation with the m6A base colored in red. (**B**) Detailed interactions between RNA backbone and CPSF-4 YTH residues. CPSF-4 YTH is shown in green cartoon or stick representation while the RNA is shown in grey or red stick representation. Dotted yellow lines display predicted direct polar contacts less than 3 Å in distance and were computed using pymol.

The online version of this article includes the following figure supplement(s) for figure 6:

**Figure supplement 1.** Structural conservation/differences of CPSF4-YTH between *Toxoplasma gondii*, *Arabidopsis thaliana* and Human YTHDC1.

shift of the polyadenylation machinery towards polyadenylation sites further downstream, thus generating elongated chimeric RNAs, which were detected by the substantial increase of read mapping at the respective distinct tandem genes. Such chimeric states of transcripts were detected recurrently in the context of KD of CPSF4 and METTL3 (*Figure 7—figure supplement 3*, *Figure 8C–D*, *Figure 8—figure supplement 2* and *Supplementary file 4*).

These chimeras displayed different patterns of alternative splicing, which can be exemplified as follows: (i) fusion transcripts covering two *loci*, each retaining the same splicing patterns as annotated, with an un-spliced, intact intergenic region (e.g. mRNA-ch1 in *Figure 8D* and mRNA-ch2 in

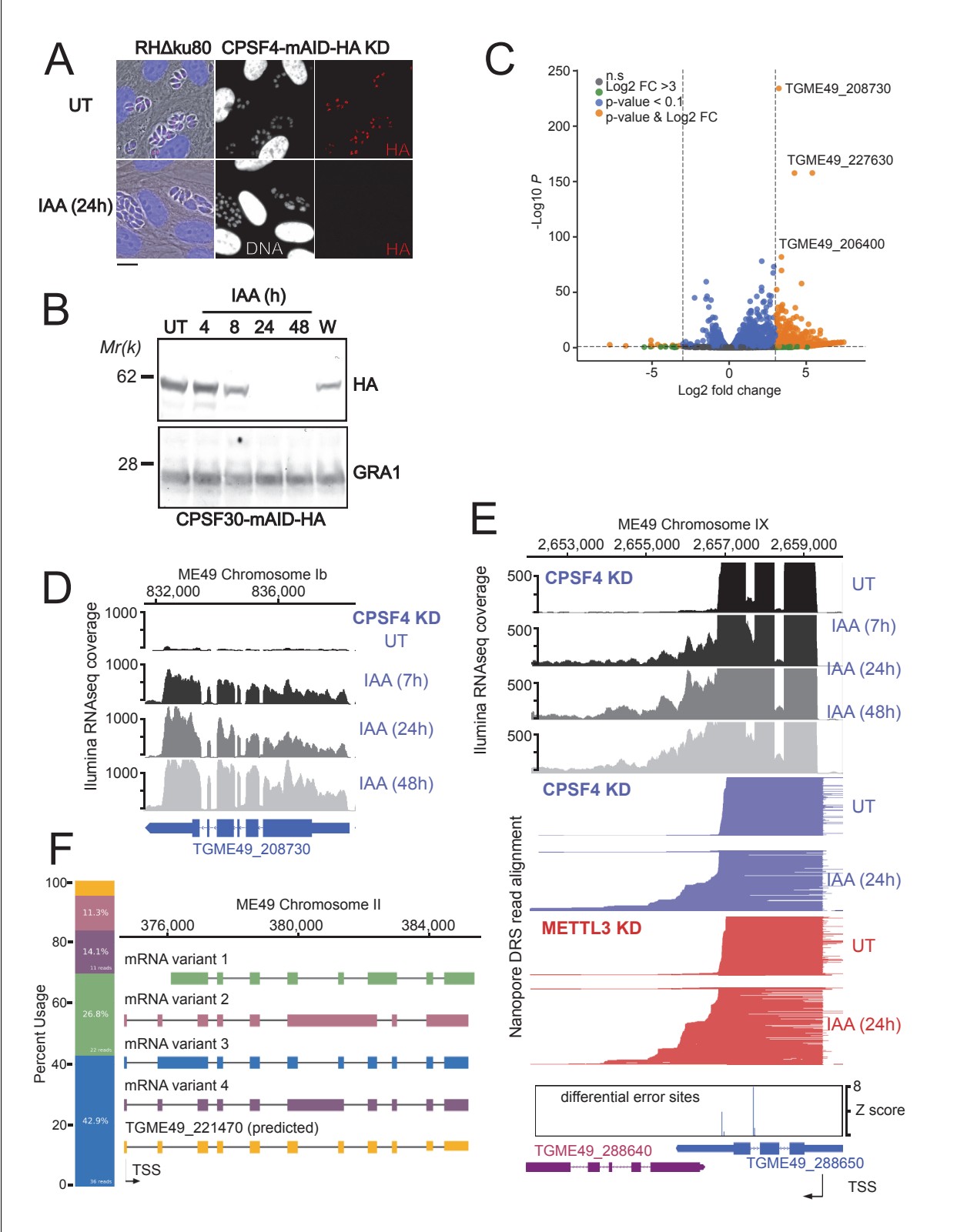

**Figure 7.** The CPSF4 post-translational Knock-down results in alternatively spliced RNAs readthrough. (**A**) CPSF4 protein expression levels after 24 hr of adding IAA, displayed by IFA on HFF cells infected with RH parasites engineered to allow the degradation of the endogenously tagged CPSF4-mAID-HA. Cells were probed with antibodies against HA (red) and DNA was stained using the Hoechst DNA dye. Scale bar, 10 μm. (**B**) Time-course analysis of the expression levels of CPSF30–mAID–HA. The samples were taken at the indicated time periods after addition of IAA and were probed with

*Figure 7 continued on next page*

*Figure 7 continued*

antibodies against HA and GRA1 (loading control). IAA-treated (24 hr) parasites were also washed (**W**), incubated with fresh media in the absence of IAA (12 hr) and analyzed using western blot. The same experiment was repeated two times and a representative blot is shown. (**C**) Volcano plot illustrating changes in RNA levels before and after the induced Knock-down of CPSF4. The orange dots indicate transcripts that were significantly up and down regulated, using adjusted p < 0.1 (Bonferroni-corrected) and ± 3-fold change as the cut-off corresponding to each comparison. X-axis showing log2 fold change, Y-axis showing -log10(p-value). Vertical dashed lines indicate threefold up- and down-regulation. (**D**) Density profile from *illumina* RNA-seq data for a representative gene targeted by the knock-down of CPSF4. RNAs were extracted from untreated cells as well as after 7, 24, and 48 hr of KD-inducing IAA treatment. RPKM values are shown on the y axis, and chromosomal positions are indicated on the x axis. (**E**) Density profile for a representative gene targeted by the KD of CPSF4, with extracted RNAs being sequenced both through *illumina*-RNA-seq data (on top) and aligned DRS reads (600 read stack for each condition, no strand coloring and no splicing characteristics displayed). The y-axis represented the read-depth. A read-through from the TGME49_288650 towards the initially repressed TGME49_288640 can be seen by both sequencing methods, following the IAA-dependent knockdown of CPSF4. A similar phenotype can be detected following the KD of METTL3. The TSS are displayed as predicted by the FLAIR isoform analysis. (**F**) FLAIR analysis was used to detect the different splicing isoforms at the *TGME49_221470* locus. The four different variants of this gene's mRNA transcripts are displayed along with their respective percentages of occurrence, on the left. Exons are shown with colored thick bars and introns with thin lines. This data was obtained from UT parasites mRNAs and was aligned against the *T. gondii* ME49 genome.

The online version of this article includes the following source data and figure supplement(s) for figure 7:

**Source data 1.** Uncropped immunoblots: Uncropped western blots corresponding to *Figure 7B*.
**Figure supplement 1.** The CPSF4 post-translational knock-down disrupts the expression levels of a multitude of genes.
**Figure supplement 2.** Read-through events and chimeric RNAs detected on *T. gondii* chromosomes V, VIIb, and XI following CPSF4 post-translational knock-down.
**Figure supplement 3.** Read-through events and chimeric RNAs detected on *T. gondii* chromosomes VIII and XII following CPSF4 and METTL3 post-translational knock-down.
**Figure supplement 4.** Nanopore-sequencing data reveals splicing complexity in *T. gondii*.

*Supplementary file 4* - panel C), (ii) fusion transcripts covering two *loci* with different splicing patterns in the intergenic region (e.g. mRNA-ch2 in *Figure 8—figure supplement 1C* and mRNA-ch7 in *Supplementary file 4* - panel E), (iii) transcripts covering two or more gene *loci* with different splicing patterns compared to the ones of the individual annotated transcript (e.g. RNA chimeras in *Figure 8—figure supplement 1C* and *Supplementary file 4* - panel E) and (iv) long transcripts covering a non-annotated region fused to an annotated transcript with variable splicing events (*Supplementary file 4* - panel C and D).

It should be noted that a few elongated transcripts showed some more complex readthrough events taking place, for instance ones involving a resulting putative collision of molecules of RNA polymerase II at opposite DNA template strands. This occasional event is exemplified by *TGME49_212260,* the transcription of which seemed to contaminate the expression of the adjacent gene *TGME49_212270*, thus forming an unusual extremely large chimeric mRNA (*Supplementary file 4* - panel G). Also, the termination defects detected following the KD of CPSF4 and METTL3 sometimes occurred at the ends of both adjacent genes generating their respective elongated chimeric transcripts that are breaching each-others transcriptional units (*Supplementary file 4* - panel H).

The unique architectural resemblance between the CPSF4 of *T. gondii* and the CPSF30L of plants (*Figure 1D* and *Figure 6—figure supplement 1C*) attracted our attention, especially in view of the recent observations made in plants of the disruption of m6A-related proteins, or CPSF30L, leading to differential polyadenylation site choices and generating longer chimeric transcripts (*Pontier et al., 2019*). This prompted us to use nanopore direct sequencing approach to accurately define the events of transcriptional readthrough in a plant model. RNAs were extracted from wild-type plants and from plants harboring either a mutation in their FIP37 gene, which encodes for a m6A methyltransferase auxiliary factor (*Shen et al., 2016*), or the *CPSF30-3* mutation (*Pontier et al., 2019*), which allows the assessment of the roles of the YTH domain, as it specifically abrogates the longer isoform of the CPSF30 gene which carries the additional YTH domain (*Figure 1—figure supplement 1D*). The nanopore DRS data generated from these samples allowed the confirmation of the existence of single full-length chimeric transcripts in the mutants, as revealed by an increase in the number of reads of a set of genes, when compared to their repressed state in the WT samples (*Figure 9A–B*). This mutation-specific increase was evidently caused by a readthrough of an upstream gene, the transcription of which did not terminate and read into the adjacent gene and terminated at the PAS of the latter instead (*Figure 9A–B* and *Supplementary file 5*). This was

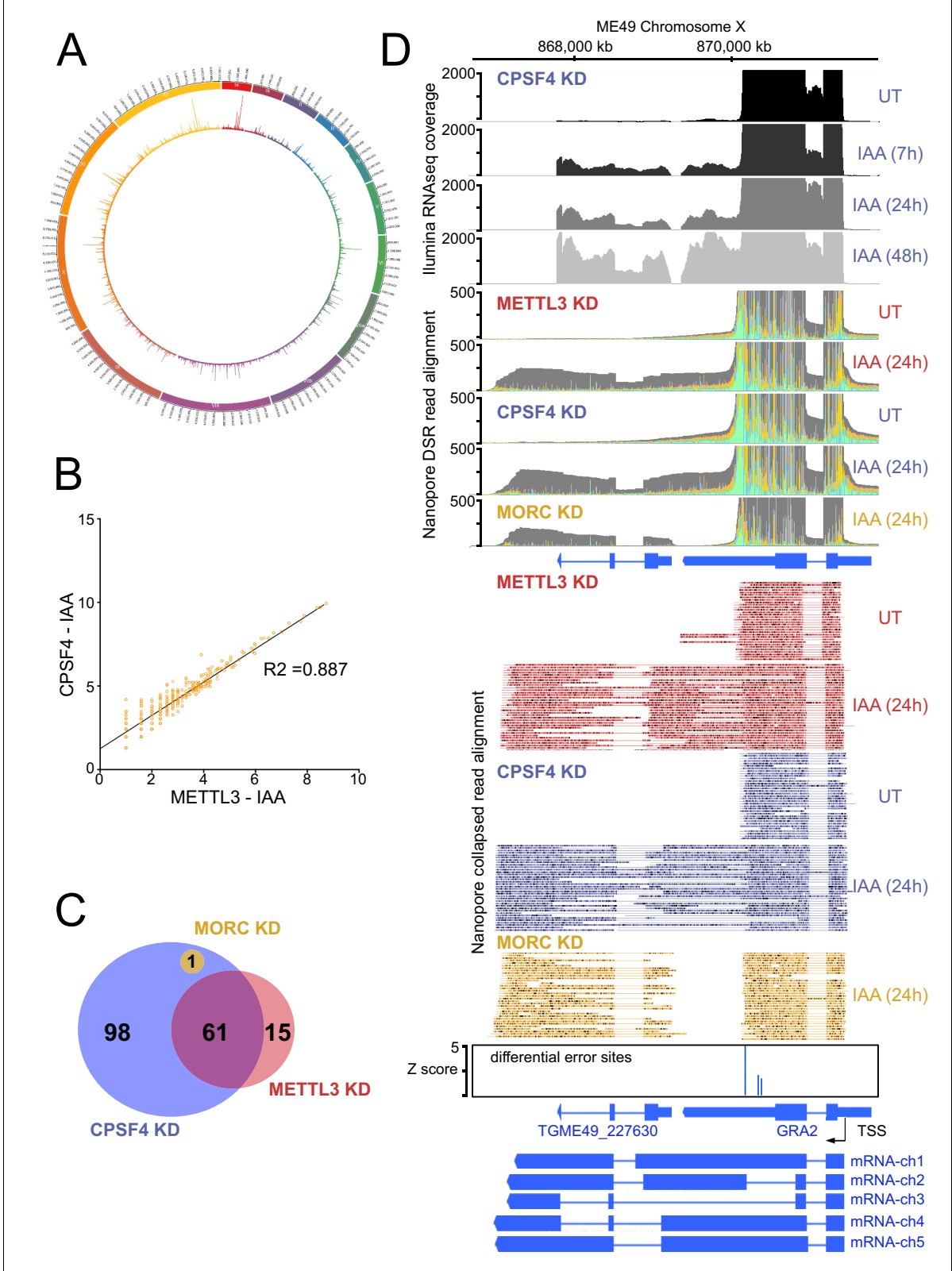

**Figure 8.** The knock-down of CPSF4, and of METTL3 generates chimeric RNAs resulting from readthrough into neighboring genes. (**A**) Circos plot displaying the assessment of average distribution of generated chimeric transcripts across the 13 *T. gondii* chromosomes, following the depletion of CPSF4 and METTL3. ChimerID scripts were used for this analysis. Histograms represent combined fold enrichment of readthroughs between UT and IAA knockdown conditions. (**B**) Scatter plot of transformed expression in two samples provided by the EdgeR analysis of chimeric reads detected by

*Figure 8 continued on next page*

*Figure 8 continued*

chimer ID comparing KD conditions of CPSF4 and METTL3. (**C**) Venn diagram of genes with an important fold change in relative chimeric read abundance (Log2 Fold change > 1, with at least three counts) within CPSF4, METTL3 or MORC KD conditions when compared to non-induced conditions. (**D**) A representative analysis of the read-through from the *GRA2* into the *TGME49_227630* locus. On top are displayed the *illumina*-RNA-seq data before and after the IAA-dependent KD of CPSF4, at different times. The y-axis represents the RPKM values. Below is the DRS aligned read multi-way pileup of RNAs extracted before and after the IAA-dependent knock-down of METTL3 (in red), of CPSF4 (in blue), and of MORC (in yellow). A more detailed close-up look is represented right below of the respective DRS data allowing a clear assessment of the read-through phenotype that is seen following the KD of both CPSF4 and METTL3. MORC KD DRS is also included to highlight a conventional upregulation of the initially repressed *TGME49_227630* gene. At the bottom is a histogram representation of the *Epinano* differential error Z-score and a schematic representation of the different RNAs expressed (readthrough differentially spliced chimeric RNAs) at these loci, based on the nanopore mapped reads following the KD of CPSF4. Exons are shown with colored thick bars and introns with thin lines.

The online version of this article includes the following source data and figure supplement(s) for figure 8:

**Source data 1.** Excel spreadsheet containing quantitative data for *Figure 8*.
**Figure supplement 1.** Read-through enrichment in CPSF4 and METTL3 KD.
**Figure supplement 2.** Read-through events and chimeric RNAs detected near *TGME49_203300* locus following CPSF4 and METTL3 post-translational knock-down.

occasionally accompanied by a differential state of the splicing of the resulting elongated transcripts. Our nanopore-based data thus provide solid proof and back up the observations of *Pontier et al., 2019* in placing the m6A machinery in plants as a safeguard against aberrant transcriptional readthrough.

## m6A-dependent polyadenylation sites, the basis of a novel mechanism of developmental gene regulation

Despite the conclusive evidence that m6A disruption generates elongated chimeric transcripts in both plants and *T. gondii*, the functional relevance of this modification in the parasite remains unknown, as does the basis for such m6A-dependent readthrough events to occur at these *loci*. The fact that the depletion of the m6A main writer enzyme, METTL3, generates transcripts that are poly-adenylated at sites downstream of their canonical ones suggests that the initial proximal PAS are ones that are dependent on the m6A modification, and that in its absence, as in the context of the METTL3 KD, a downstream m6A-independent PAS is chosen by the polyadenylation machinery, thus generating the poly(A) transcripts that are detected by nanopore mRNA sequencing.

To support this claim, and to identify the sites of m6A across the genome, we employed DRS to indirectly detect modified nucleotides. In fact, the presence of an m6A mark is known to induce a higher rate of base calling errors on, or within, the close proximity of m6A sites, as would any other nucleotide modification. The recently developed *Epinano* pipeline (*Liu et al., 2019*) uses error variations between two sets of aligned DRS reads (WT vs KD) to map significantly modified error sites. Using such an approach to compare UT vs IAA-dependent METTL3-KD DRS datasets, allowed us to presume that most of these detected differential error sites are m6A sites (*Figure 10A*). We then analyzed the motifs around which the most significant peaks of error were mapped, which revealed a high and significant enrichment of a motif consisting of ARACW (R = A/G, W = A/T/G) (*Figure 10B*). This resembles the RGAC core motif which is the established m6A consensus sequence identified in *P. falciparum* (*Baumgarten et al., 2019*), *A. thaliana* (*Parker et al., 2020*), humans (*Linder et al., 2015*), and yeast (*Schwartz et al., 2013*). We were able to also confirm the m6A signature that was identified in *Arabidopsis* (*Parker et al., 2020*) using our nanopore data (*Figure 10B*). About 65% of the error sites mapped at the RRAC consensus motif, which seems to be evolutionary conserved across canonical strains of *T. gondii*, as shown by the evaluation of individual methylation sites, suggesting that mRNA methylation is a *cis*-regulatory feature conserved at the gene level (*Figure 10D–F* and *Figure 10—figure supplement 1B*).

This error-based identification of methylation sites enabled us to locate putative m6A sites mostly at 3'UTR (*Figure 10C*), but most importantly, at sites where the canonical proximal PAS is overrun, as seen in the context of the depletion of METTL3 and CPSF4 (*Figure 10D–F*, *Figure 10—figure supplement 1B*, *Figure 7—figure supplement 2B*, *Figure 7—figure supplement 3*, *Figure 8D*, and *Supplementary file 4*). Thus, we believe that the choice of this site is initially regulated by the m6A modification. In fact, many loci with termination defects presented significant error peaks at the

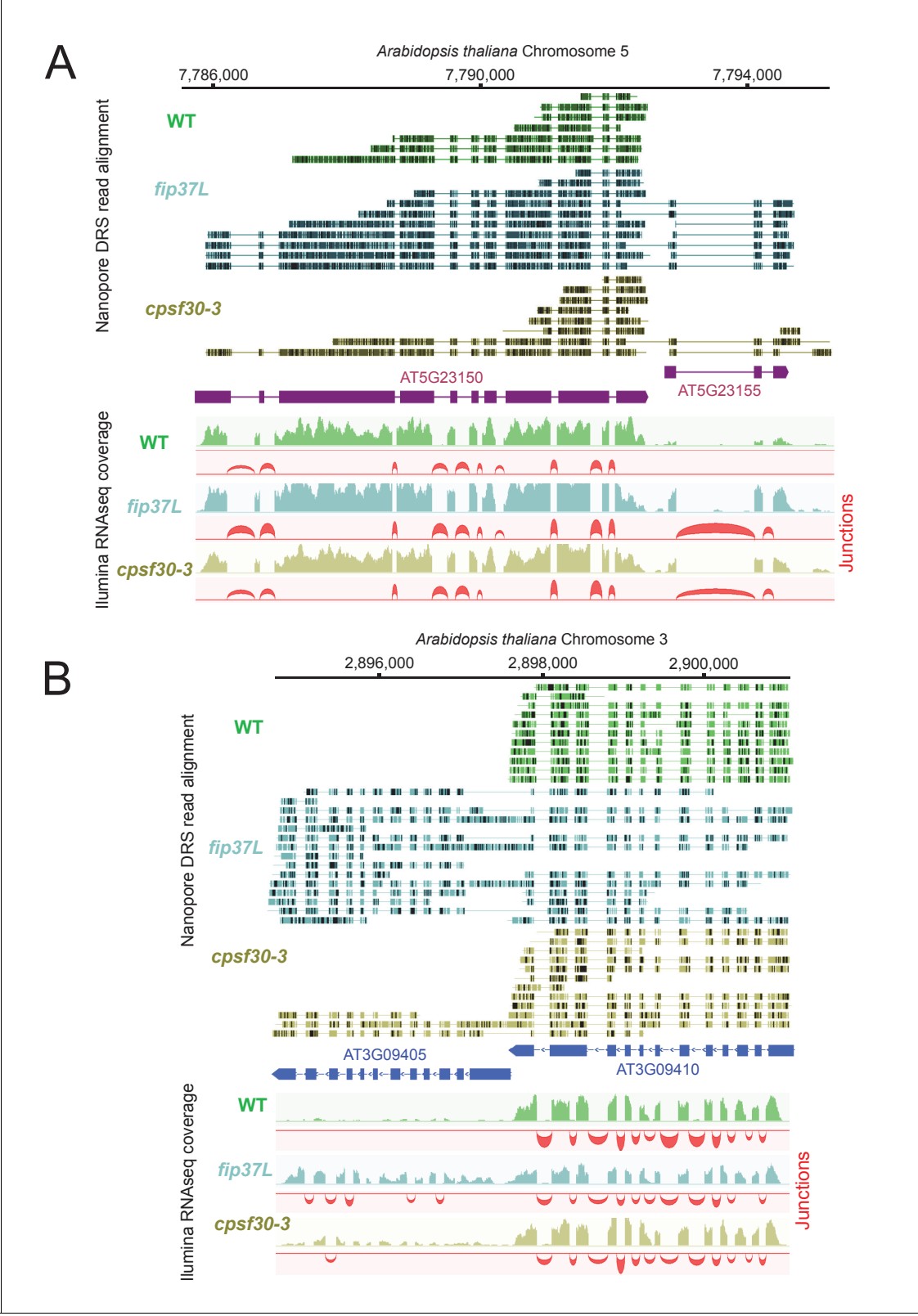

**Figure 9.** The CPSF4 homologue in plants (CPSF30L), similarly to *T. gondii*, prevents the formation of RNAs readthrough. (**A**) DRS aligned reads (top) and illumina RNA-seq density plots (bottom) of the *AT5G23150-AT5G23155* loci displaying a readthrough when the RNAs are extracted from plants harboring either the *fip37L* mutation (Fip37 is an m6A methyltransferase auxiliary factor), or the *CPSF30-3* mutation which specifically abrogates the CPSF30L mRNA production (see ***Figure 1—figure supplement 1D***) thus allowing an assessment of the roles of the YTH domain. The *illumina* RNA-seq

*Figure 9 continued on next page*

*Figure 9 continued*

data are represented by sashimi plots showing the differential splicing outcomes on the introns, in the backgrounds of these mutations. (**B**) Density profiles from both Nanopore RNA-sequencing (top) and illumina RNA-seq data (bottom) of the *AT3G09410 - AT3G09405* loci. Similar description as in (**A**).

putative 3' termination site (*Figure 10C*). This could be explained by an overlap existing initially between the overrun PAS sites and m6A sites, if not that the adenosines of the PAS themselves could be methylated, as was observed in plants (*Parker et al., 2020*). We could not, however, show a statistical relation between the presence of an error site and the obligate formation of a read-through as many transcripts also have proper termination at the 3'end.

We believe that, in the natural WT case, the m6A site would guide the polyadenylation machinery, via the ability of the CPSF4 YTH to bind this modification, thus allowing the recognition of the respective proximal PAS site, and the proper termination at this *locus*. The phenomenon observed is, however, probably under representative of the biological reality as we can expect many abortive transcripts to end either non poly-adenylated, or degraded, rendering the poly-A dependent DRS impossible. These observations thus argue in favor of the existence of both m6A-dependent and m6A-independent PAS, these respectively being represented by the proximal and the distal/down-stream PAS. This pushed us to tackle the nature of the genes within the *loci* that are exhibiting the depicted readthrough, and displaying these double PAS features.

The readthrough of the gene upstream, hereafter referred to as gene1, invading the transcriptional unit of the gene downstream, hereafter referred to as gene2, seemed to be occurring at *loci* exhibiting a distinctive pattern. In fact, the genes two that were displaying now higher amounts of nanopore-reads in the context of KD of METTL3 and CPSF4, were mostly, if not all, initially repressed, and represented developmentally regulated genes, that happen to be adjacent to expressed tachyzoite genes. The mRNA analysis of the set of genes that were targeted by this read-through phenotype, illustrated their developmentally regulated nature (*Figure 11A*). Interestingly, many of these genes are recognized as targets of the MORC repressor complex and their expression is seen to be upregulated following the KD of MORC (*Figure 11A*; *Farhat et al., 2020*). However, a detailed look at the nanopore-derived reads in the contexts of KD of the latter, when compared with those of CPSF4 and METTL3, provided enough proof that the read-through phenotype occurs following the KD of both CPSF4 or METTL3, but not of MORC, which only resulted in a conventional promoter-dependent upregulation of the initially repressed genes (*Figure 8D*, *Figure 10—figure supplement 1B*, *Figure 11B–D* and *Supplementary file 4*). Apart from serving as a control arguing in favor of the specificity of the depicted formation of chimeric RNAs (*Figure 11D*), the data generated in the context of MORC KD helped distinguishing the mis-annotation of certain genes, such as the example shown for the *TgME49_227630* (*Figure 8D*), avoiding any misinterpretation of the elongated transcripts.

The recurrence of the dual expression pattern between gene1 and gene2, the first being specific to tachyzoite, and the second being repressed and only expressed in stages other than tachyzoite, suggests that the respective m6A-dependent polyadenylation of gene1 and the m6A independent polyadenylation of gene2, at the core of an essential mechanism aiding in the tight transcriptional regulation of developmental stage-specific regulated genes in *T. gondii*.

An illustrative example of this observation can be that of *ROP35* (*Figure 11B–D*), a rhoptry gene that displays a tachyzoites specific expression, and which occurs upstream of a repressed gene, namely *TgME49_304730*, the expression of which is known to be specific to the late sexual, early oocyst stages (EES5 and oocyst D0) (*Figure 11A*). The mRNA levels of *ROP35* were unaltered following the KD of METTL3 or of CPSF4, based on both *illumina*-seq and nanopore-seq data. However, the expression of the downstream *TgME49_304730* was clearly induced following the KD of CPSF4, as illustrated by *illumina*-seq (*Figure 11B*, top). Similarly, DRS displayed a higher level of reads at this *locus* and in the context of KD (*Figure 11B*, bottom). The analysis of the reads generated at these *loci* provides the evidence for this upregulation to be caused by an overrun of the *ROP35* PAS and the readthrough breaching the transcriptional unit of the downstream gene2 (*TgME49_304730*), as well as the polyadenylation machinery terminating by using this alternative PAS, which can now be referred to as an m6A-independent PAS, as evidenced by the earlier results and the error-based m6A site identification (*Figure 11C*).

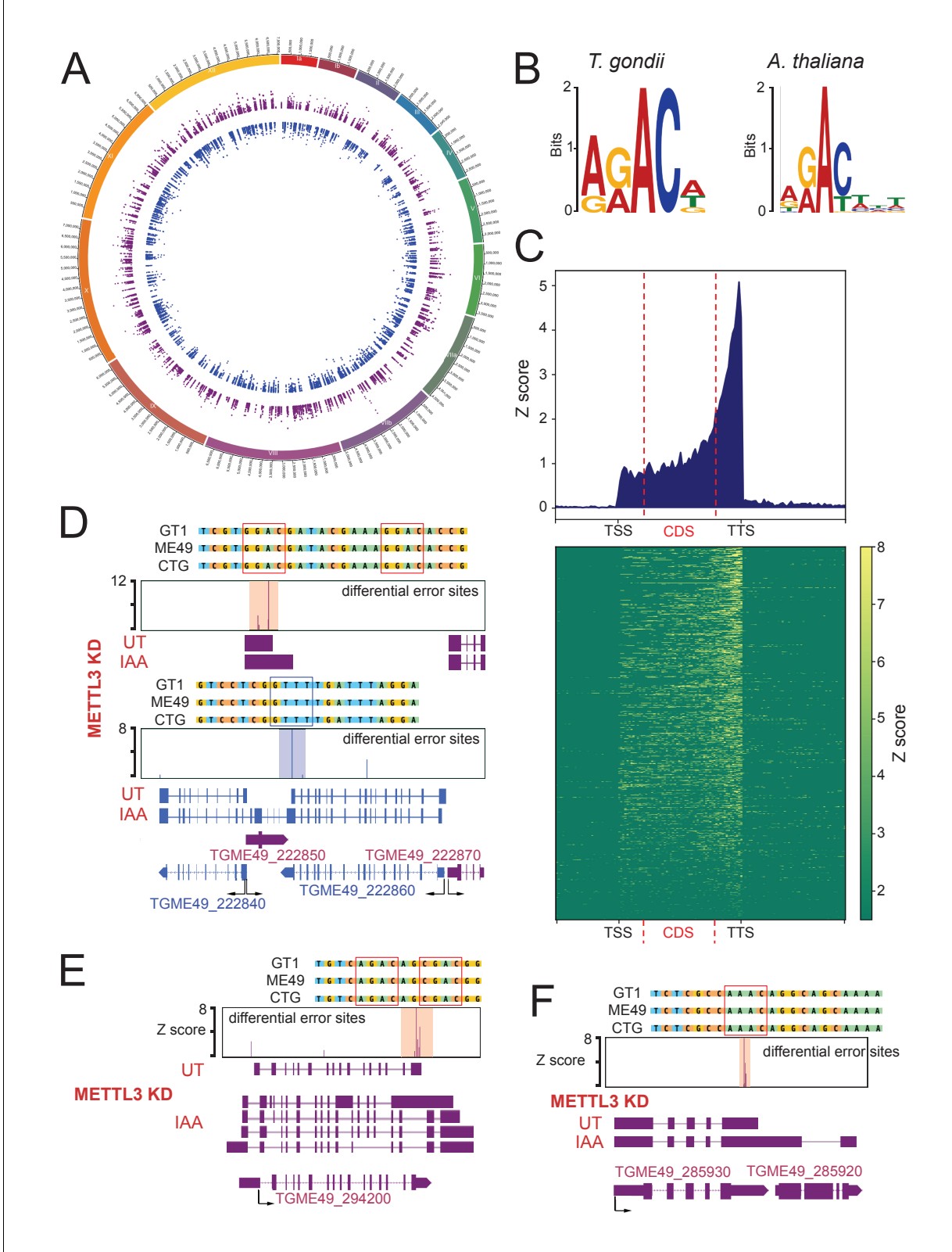

**Figure 10.** Differential error rate analysis identifies sites of METTL3-dependent m6A modifications transcriptome-wide in *T. gondii*. (**A**) Circos plot showing the distribution across the 13 *T. gondii* chromosomes of the m6A sites predicted based on peaks corresponding to differential error rates determined after depletion of METTL3. (**B**) The motif at error rate sites matches the consensus m6A target sequence. The sequence logo is for the motif enriched at sites with differential error rate in *T. gondii* METTL3 KD (left logo) and *A. thaliana fip37L* mutant (right logo) (**C**) Metagene plot

*Figure 10 continued on next page*

*Figure 10 continued*

showing average Z-score of differential error sites along a normalized transcript with a clustered Z-score heatmap displaying individual transcripts with a sufficient depth for analysis (>50 reads). The m6A mark is most abundant in annotated 3'-UTRs. (D) Detection by nanopore of a large chimeric mRNA originated from the readthrough of *TGME49_222860* into the *TGME49_222840* locus and on the opposite strand the impaired termination at *TGME49_222850*. The nanopore differential error sites and the alignment of sequences containing the m6A motif of *T. gondii* strains are shown. (E) Schematic representation of the different RNAs expressed at the *TGME49_294200* locus upon METTL3 depletion, based on nanopore-mapped reads. Exons are shown with colored thick bars and introns with thin lines. The read-through beyond the locus boundaries is displayed, as well as the differential splicing outcomes on the respective exons, resulting in chimeric RNAs. (F) Schematic representation of the nanopore mapped RNA reads illustrating the readthrough from *TGME49_285930* into the *TGME49_285920* locus following the KD of METTL3. Exons are shown with colored thick bars and introns with thin lines. The nanopore differential error sites (Z-score >50) are indicated and a zoomed peak-centered view shows the sequence containing m6A consensus motif across the three canonical strains of *T. gondii*.

The online version of this article includes the following figure supplement(s) for figure 10:

**Figure supplement 1.** Read-through events and chimeric RNAs detected near *TGME49_244700* locus following CPSF4 and METTL3 post-translational knock-down.

## Discussion

The correct processing of the 5' and 3' ends of mRNA is paramount to the effective expression of any functional gene for all eukaryotes. In apicomplexan parasites, 3' end processing, and notably cleavage and polyadenylation, are emerging as attractive targets for chemical inhibition (*Bellini et al., 2020*; *Palencia et al., 2017*; *Swale et al., 2019*) as these highly replicative cells strongly depend on consistent mRNA production. Here, using biochemistry, we describe in detail the components of the CPSF core complex within *T. gondii*. Although not an abundant complex, we observe conservation in component architecture, uncovering most of the described CPSF subunit orthologs in higher mammals (including CPSF1, CPSF2, CPSF3, CPSF4, SYMPLEKIN, and WDR33) together with the associated CSTF factors and Fip1. Altogether, with PAP, which is purified mostly as a single module, we obtained the necessary components for PAS signal recognition, cleavage, and polyadenylation. Although conserved in composition, the subunits themselves display important sequence divergence and are rarely show greater than 30% sequence identity. These proteins are all encoded by essential genes, highlighting the strict dependency on a fully functional CPSF complex. Furthermore, some of the identified core subunits were difficult to trace back to a corresponding ortholog in mammals, further suggesting potential divergence in function when compared to other organisms.

Among these subunits, CPSF4, a crucial subunit of the PAS recognition complex, shows a bipartite divergence with strong functional implications in *T. gondii*. First, the N-terminal zinc finger domains, essential units in the canonical PAS motif (AUAAA) interaction, show only a partial conservation, which leads us to speculate that although their function in PAS motif binding is probably conserved, the recognized motif may have diverged. The second more striking element is the presence of an additional C-terminal YTH domain, which implies a direct linkage to the m6A mark. This evolutionary feature is found the *Apicomplexa* ancestor *Chromera velia*, but also in the more distantly related plant phyla, and highlights a potential functional convergence in the mechanisms of PAS recognition and interaction in these species.

We have shown, using ITC and high-resolution crystallographic structures, that this domain is indeed functional and exclusively binds m6A-modified RNAs, consistent with its putative predicted function. The deep hydrophobic pocket dedicated to m6A recognition is for the most part similar to other YTH domains, with the noticeable difference attributed to a supplemental small α-helice (α3) which opens the site and makes it comparatively more accessible than the HsYTHDC1 or AtCPSF30-YTH structures (*Hou et al., 2021*). This feature can explain the quantitatively weaker interaction when compared to AtCPSF30-YTH or YTHDC1 affinities which are within the nanomolar range. The binding of RNA occurs in a similar fashion to other YTH/RNA structures (YTHDC1 and AtCPSF30-YTH) and no elements forcing sequence specificity are clearly visible in this structure as most of the bases are turned outwards and only the phosphate backbone interacts with the negatively charged binding groove. Intriguingly, however, the presence of a potential secondary binding groove implies possibly multiple binding modes which could depend on RNA secondary structures or on the binding context with the PAS. This feature, combined with a lower affinity, is indicative of a high plasticity in RNA binding with the only centerpiece being the presence or absence of m6A. The YTH domain

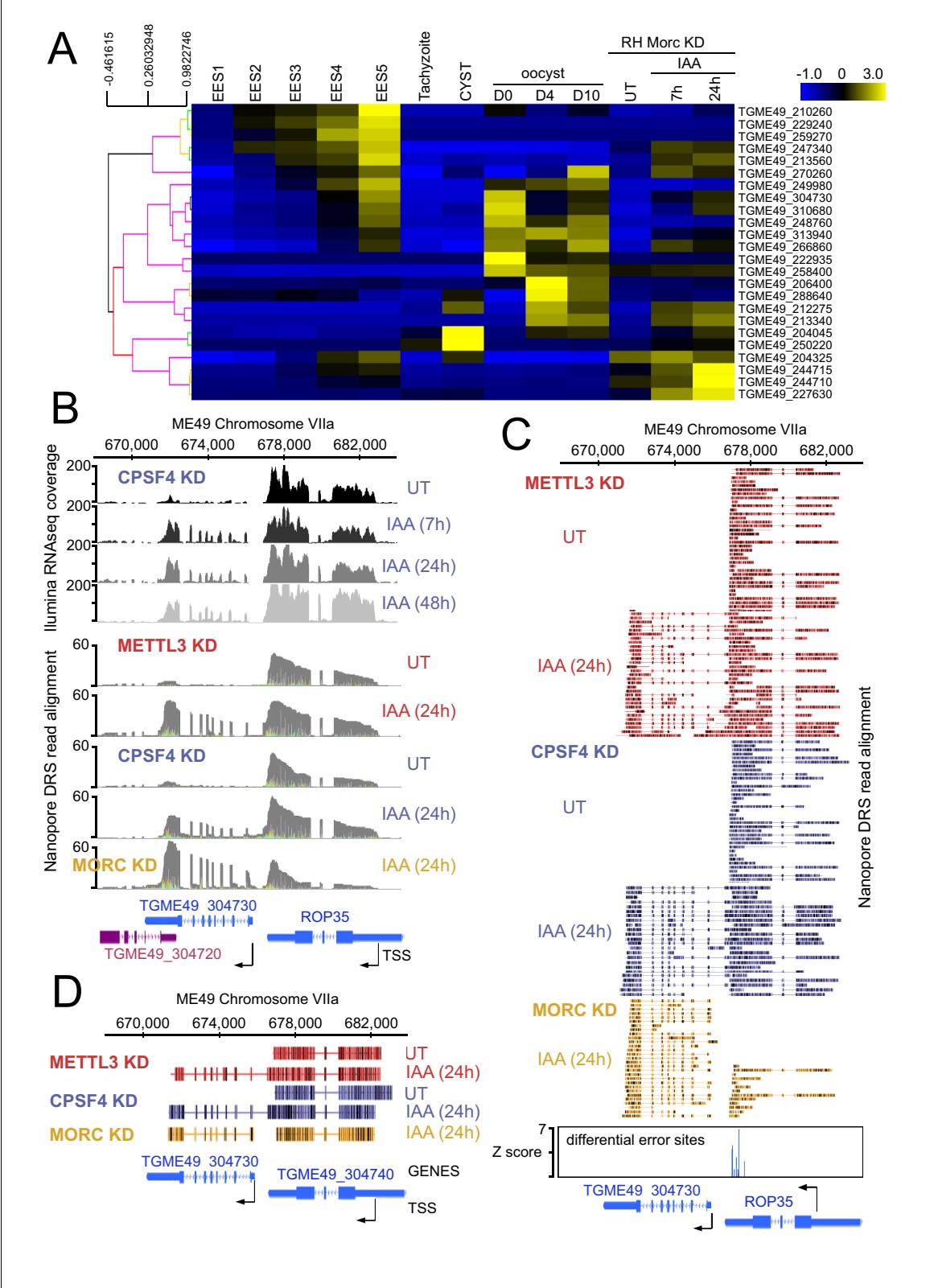

**Figure 11.** CPSF4 and METTL3 both acts to prevent the readthrough into developmentally regulated genes. (**A**) A heat map representation showing mRNA hierarchical clustering analysis (Pearson correlation) of a set of genes targeted by the readthrough phenotype following the KD of CPSF4 and METTL3, and which have been already established to be upregulated following the KD of MORC. Displayed are the abundance of their respective transcripts before and after the depletion of MORC (*Farhat et al., 2020*), as well as during the different life cycle stages the data of which are collected

*Figure 11 continued on next page*

*Figure 11 continued*

from ToxoDB published transcriptomes of merozoite, longitudinal studies on enteroepithelial stages (EES1 to EES5), tachyzoites, bradyzoites, and cysts from both acute and chronically infected mice, and finally of immature (day 0), maturing (day 4) and mature (day 10) stages of oocyst development. The color scale indicates log2-transformed fold changes. (**B**) A representative analysis of the read-through from the *ROP35* transcript into the *TGME49_304730* locus. On top are displayed the illumina-RNA-seq data before and after the IAA-dependent KD of CPSF4, at different times. The y-axis represents the RPKM values. Below is the nanopore-based RNA sequencing of RNAs extracted before and after the IAA-dependent knock-down of METTL3 (in red), of CPSF4 (in blue), and of MORC (in yellow). The y-axis represents the read-depth counts. (**C**) A more detailed close-up look of the respective nanopore data at the same *locus* allowing a clear assessment of the read-through phenotype that is seen following the KD of both CPSF4 and METTL3 but not of MORC which only resulted in a conventional upregulation of the initially repressed *TGME49_304730* gene. The accuracy of the nanopore data is seen here with the relative repression of the *ROP35* gene following the KD of MORC, as seen in the illumina RNA-seq data (*Farhat et al., 2020*). The number of the mapped reads was adjusted between the data from the different experimental conditions. (**D**) A schematic representation of the nanopore mapped RNA reads illustrating the readthrough from *ROP35* into the *TGME49_304730* locus following the KD of CPSF4 and of METTL3, but not after the KD of MORC which only resulted in a conventional transcriptional upregulation of the initially repressed TGME49_304730 gene. Exons are shown with colored thick bars and introns with thin lines.

is separated from the zinc fingers by an extended region which is predicted as disordered. This observation highlights the flexible nature of CPSF4 which itself integrates the quaternary CPSF complex. Recently, *Song et al., 2021* also showed that this same disordered region was involved in promoting liquid-liquid phase separated organelles by the CPSF30-L of *Arabidopsis thaliana*. Further structural research may uncover the interplay between m6A recognition and PAS recognition in the context of a fully active CPSF complex.

In parallel to the orthodox 3'end processing mechanism which occurs independently of the m6A modification, we propose the existence in *T. gondii* of an m6A-dependent polyadenylation through which the m6A site would guide the polyadenylation machinery, via the ability of the CPSF4 YTH to bind this modification, thus allowing the recognition of the respective m6A-dependent polyadenylation site, and the proper termination at a relevant *locus*.

The fact that the parasite has adopted and evolved such an unconventional 3'end processing mechanism, suggests a role in the tight regulation of gene expression that is typical of this highly adaptive organism. This m6A dependent polyadenylation was detected mainly at the ends of a set of tachyzoites-specific genes that are highly expressed and are adjacent to developmental stage-specific repressed genes (*Farhat et al., 2020*), hence at sites where a highly efficient barrier is needed to partition the distinct transcriptional signatures of these tandem genes, thus preventing any aberrant readthrough of the polymerase that is actively transcribing the upstream tachyzoite gene.

When taking into consideration the pervasive nature of transcription in the highly replicative tachyzoites stage, along with the remarkably high level of gene density of this parasite's genome, which bears very few constitutive heterochromatic regions, the relevance of the parasite adopting additional means for preventing any aberrant transcription of its repressed genes at the tachyzoite stage becomes clear. The high rate of transcription that is witnessed in this stage, dampens down in the other stages, these later stages being either slow in their proliferation or even quiescent, which might explain why the parasite has prioritized a large set of tachyzoite genes by this m6A related transcriptional barrier at their 3'ends. Additionally, the fact that most of the m6A related enzymes and most of the 3'end processing factors, were found to be less expressed in latent stages than they were in highly replicative ones, agrees with the requirement of this mark at these stages in particular (*Figure 1—figure supplement 1C* and *Figure 2—figure supplement 1D*).

Although this differential concentration might hint at some level of stage-specific upstream regulation for this mark, the fact that no m6A erasers have been detected in *T. gondii* and that the mark seems to have a relatively long half-life hints at a low level of dynamism for this mark. It seems that the crucial requirement for this m6A-dependent barrier in tachyzoites might not be extended to other stages. When transcribed, the transcriptional termination of the transcripts of genes 2 (as referred to the text to the downstream gene in a tandem, which belongs to stages other than tachyzoites) would occur in an orthodox manner, independently of m6A.

It should be noted that the PAS sites which we referred to as being m6A-dependent were not sequenced, thus we do not claim that *T. gondii* harbors the canonical AAUAAA. However, in metazoans, variants of this hexamer have been observed (*Shepard et al., 2011*), and at plants transcripts,

this sequence does not represent more than 10% of the total PAS (*Loke et al., 2005*), and in other species, auxiliary cis-elements were reported to be substituting for the lack of any PAS elements at some transcripts, in their transcripts 3'end processing purpose (*Nunes et al., 2010*; *Venkataraman et al., 2005*).

In *Arabidopsis thaliana*, many sequenced PAS were classified as being m6A-dependent, with the termination of the respective transcripts being dependent on the specific binding of this modification by the YTH within CPSF30L (*Parker et al., 2020*; *Pontier et al., 2019*). However, the existence of more than 12 YTH-proteins in plants, hints at the involvement of the m6A mark in a multitude of mechanisms (*Figure 2—figure supplement 1A*). The magnitude of transposable elements is acknowledged in plants species, as they represent driving evolutionary forces of gene expansion and duplication, thus of genome complexity in these species (*Rensing, 2014*). Despite these elements aiding the static plants in their constant requirement for adaptation to their changing environment, their mis-regulation and aberrant transcription could generate detrimental regulatory effects, thereby requiring additional means to keep these elements in check. The m6A-assisted polyadenylation has recently been suggested to serve as one of the mechanisms for prevention of aberrant transcription at recently rearranged *loci* (*Pontier et al., 2019*). Despite the functional conservation of the architecture of CPSF4 between plants and *T. gondii*, the evolutionary divergence between these species and the different homeostatic requirements emanating from either a static but free life style, or a multi-host parasitic one, drove each of them to evolve this m6A-dependent barrier for their respective needs. In *T. gondii*, this barrier would answer to one of the most discernible challenges that the parasite faces: its need to partition the distinct stage specific transcriptional signatures of its genes which in the majority of cases have their transcriptional units bordering each other, if not even overlapping.

In fact, if it were not for such a high level of gene density occurring in the genome of *T. gondii*, the traversal and overrun of a gene's PAS, without the conventional downstream RNA cleavage and polyadenylation would have led in most cases to an aberrant non-adenylated and eventually degraded transcripts. The close distance between the adjacent genes makes it so that the RNA pol II would still be able to scan the downstream poly(A) signal site and to use it to efficiently terminate the transcripts, thus allowing us to detect these latter by nanopore and to assess the phenotype of the KD of CPSF4 at these *loci*.

In addition, employing the nanopore DRS allowed us to observe events which we could not have captured through *illumina*-seq. Apart from the genome assembly artefacts emanating from the inaccuracy of conventional techniques to read the repetitive elements which exist broadly in this genome, there seem to be a fairly large amount of *loci* that were mis-annotated or even non-annotated in the genome of *T. gondii*. For instance, nanopore-based DRS allowed us to align the transcripts of some unannotated genes (e.g. *Supplementary file 4* – panel D). The nanopore data also allowed us to distinguish the direction in which the transcription is taking place; for instance, the readthrough breaching into the *TGME49_212275* gene (*Supplementary file 4* – panel G) occurs in a direction opposite to the strand at which it is predicted, in the context of the CPSF4-KD-dependent readthrough, while the transcription of this same gene follows its predicted direction in the context of its MORC-KD-dependent induction. A similar behavior is observed in *Figure 10—figure supplement 1*. The ability to detect the orientation of the transcription occurring at a gene, allowed us to also predict instances of steric hindrance between molecules of actively transcribing polymerases, as the case at a gene that was initially transcribed in WT, but then had fewer reads mapping at its *locus* when the adjacent repressed gene was undergoing a CPSF4-KD-dependent readthrough (*Figure 7—figure supplement 2D*).

Despite witnessing a clear-cut loss of a crucial post-transcriptional barrier following the disruption of either the deposition or the reading of m6A, it cannot be excluded that the genomic context could have had an impact on the degree of the transcriptional functional outcomes observed at those *loci*. Also, this modification has been linked to the translational potential of transcripts as well as to their stability in other apicomplexan parasites, namely in *Plasmodium falciparum* (*Baumgarten et al., 2019*), which brings our attention to whether this m6A/polyadenylation coupling would also serve this parasite in its transcriptional partitioning, especially in view of the conservation of the special evolutionary feature of a YTH being carried within its CPSF4 homolog.

# Materials and methods

## Parasites and human cell culture

HFFs (ATCC CCL-171) were cultured in Dulbecco's modified Eagle medium (DMEM; Invitrogen) supplemented with 10% heat inactivated fetal bovine serum (Invitrogen), 10 mM HEPES buffer pH 7.2, 2 mM l-glutamine and 50 µg ml−1 of penicillin and streptomycin (Invitrogen). Cells were incubated at 37°C in 5% CO. The *Toxoplasma* strains that were used in this study (listed in *Supplementary file 6*) were maintained in vitro by serial passage on monolayers of HFFs. The cultures were free of mycoplasma, as determined by qualitative PCR.

## Endogenous tagging of CPSF, WTAP, and METTL subunits

A list of the plasmids and primers for genes of interest (GOIs) that were used in this study is provided in *Supplementary file 6*. To construct the vector pLIC-GOI-HAFlag, the coding sequence of GOI was amplified with the primers LIC-GOI-Fwd and LIC-GOI-Rev, using *T. gondii* genomic DNA as a template. The resulting PCR product was cloned into the pLIC-HA-Flag-dhfr or pLIC-(TY)2-hxgprt vectors using the ligation-independent cloning method. The plasmid pTOXO_Cas9-CRISPR was described previously. We cloned 20-mer oligonucleotides corresponding to specific GOIs using the Golden Gate strategy. In brief, primers GOI-gRNA-Fwd and GOI-gRNA-Rev containing the sgRNA targeting GOI genomic sequence were phosphorylated, annealed and ligated into the pTOXO_-Cas9-CRISPR plasmid linearized with BsaI, leading to pTOXO_Cas9-CRISPR-sgGOI. The same approach was also used to build pLIC-GOI-HAFlag-mAID vectors as already described in *Farhat et al., 2020*.

## Transfection of *T. gondii*

*T. gondii* strains were electroporated with vectors in cytomix buffer (120 mM KCl, 0.15 mM CaCl, 10 mM K$_2$HPO$_4$, 10 mM KH$_2$PO$_4$ pH 7.6, 25 mM HEPES pH 7.6, 2 mM EGTA, and 5 mM MgCl$_2$) using a BTX ECM 630 machine (Harvard Apparatus). Electroporation was performed in a 2 mm cuvette at 1.100 V, 25 Ω and 25 µF. When needed, the following antibiotics were used for drug selection: chloramphenicol (20 µM), mycophenolic acid (25 µg ml−1) with xanthine (50 µg ml−1), pyrimethamine (3 µM), or 5-fluorodeoxyuracil (10 µM). Stable transgenic parasites were selected with the appropriate antibiotic, single- cloned in 96-well plates by limiting dilution, and verified by immunofluorescence assay or genomic analysis.

## Reagents

The following primary antibodies were used in the immunofluorescence, mouse anti-HA tag (Roche; RRID, AB_2314622), rabbit anti-HA Tag (Cell Signaling Technology; RRID, AB_1549585), rabbit anti-N6-methyladenosine (m6A) (Merck Millipore), rabbit anti-TgGAP45 and mouse anti-Ty (gifts from D. Soldati, University of Geneva). Immunofluorescence secondary antibodies were coupled with Alexa Fluor 488 or Alexa Fluor 594 (Thermo Fisher Scientific).

## Endogenously tagged protein immunoprecipitation

*T. gondii* extracts from RHΔku80 cells stably expressing HAFlag-tagged CPSF1, Fip1, PAP, WDR33, WTAP, METTL3 or METTL14 were incubated with anti-FLAG M2 affinity gel (Sigma-Aldrich) for 1 hr at 4°C. Beads were washed with 10-column volumes of BC500 buffer (20 mM Tris-HCl, pH 8.0, 500 mM KCl, 20% glycerol, 1 mM EDTA, 1 mM DTT, 0.5% NP-40, and protease inhibitors). Bound polypeptides were eluted stepwise with 250 µg/ml FLAG peptide (Sigma Aldrich) diluted in BC100 buffer. After the identification of elution fractions by SDS-PAGE using silver nitrate revelation (Sigma Aldrich), fractions of interest were precipitated in trichloroacetic acid for 1 hr at 4°C, centrifuged for 30 min at 14,000 g and dried overnight at room temperature. For MS profiling, the entire precipitate was resuspended in 1X SDS-PAGE loading buffer, heated 5 min at 90°C then loaded on NuPAGE gels.

## MS-based interactome analyses

Protein bands were excised from gels stained with colloidal blue (Thermo Fisher Scientific) before in-gel digestion using modified trypsin (Promega, sequencing grade). The resulting peptides were

analyzed by online nanoliquid chromatography coupled to tandem mass spectrometry (UltiMate 3000 and LTQ- Orbitrap Velos Pro (Thermo Fisher Scientific) for the CPSF1, WDR33, FIP1 et PAP interactomes, and UltiMate 3000 RSLCnano and Q-Exactive Plus (Thermo Fisher Scientific) for the METTL3, METTL14 and WTAP interactomes). The peptides were sampled on a 300 μm × 5 mm Reprosil-Pur 120 C18-AQ, 5 μm, C18 precolumn and separated on a 75 μm × 250 mm Reprosil-Pur 120 C18-AQ, 1.9 μm, column (Dr. Maisch) using gradients of 25 min for the CPSF1, WDR33, FIP1 et PAP interactomes. For the METTL3, METTL14 and WTAP interactomes, the peptides were sampled on a 300 μm x 5 mm precolumn (PepMap C18, Thermo Scientific) and separated on a 75 μm x 250 mm column (Reprosil-Pur 120 C18-AQ, 1.9 μm, Dr. Maisch) using gradients of 35 min. MS and MS/MS data were acquired using Xcalibur (Thermo Fisher Scientific). Peptides and proteins were identified using Mascot (v.2.7.0) through concomitant searches against the *Toxoplasma gondii* database (ME49 taxonomy, v.30, downloaded from ToxoDB), the UniProt database (*H. sapiens* taxonomy, January 2021 version), a homemade database containing the sequences of 250 classical contaminants found in proteomic analyses (human keratins, trypsin, bovine albumin, etc.), and the corresponding reversed databases. Trypsin was chosen as the enzyme and two missed cleavages were allowed. Precursor and fragment mass error tolerances were respectively set at 10 ppm and 0.6 Da for LTQ-Orbitrap Velos Pro data and 10 ppm and 20 ppm for Q-Exactive Plus data. Peptide modifications allowed during the search were: carbamidomethyl (C, fixed), acetyl (protein N-term, variable) and oxidation (M, variable). The Proline software was then used to perform a compilation and grouping of the protein groups identified in the different bands. Proline was also used to filter the results: conservation of rank one peptide spectrum matches (PSMs), peptide length ≥ six amino acids, PSM score ≥ 25, false discovery rate of peptide-spectrum-match identifications < 1% as calculated on peptide-spectrum-match scores by employing the reverse database strategy, and a minimum of 1 specific peptide per identified protein group. Proteins from the contaminant database were discarded from the final list of identified proteins. MS1-based label-free quantification of the protein groups was performed using Proline to infer intensity-based absolute quantification values (iBAQ) that were used to rank identified *Toxoplasma* proteins in the interactomes. The MS proteomics data have been deposited to the ProteomeXchange Consortium through the PRIDE partner repository with the dataset identifier PXD024326.

## Recombinant expression of TgCPSF4 YTH

TgCPSF4 (277–445) was codon optimized for *E. coli*, synthetized and cloned by Genscript within a modified pET30-a (+) vector (Addgene) in order to possess an N-Terminal, TEV cleavable, 8*His Tag. Expression of the recombinant protein was performed in BL21(DE3) chemically competent cells. Briefly, on day 1, 50 μl of BL-21(DE3) cells were incubated with 1 ug of plasmid for 10 min at 4°C, transformed by heat shock at 42°C for 45 s and further incubated 10 min on ice. Following transformation, 600 μl of Luria Broth (LB - Formedium) were added and a 1 hr, 37°C pre-culture was undertaken before platting 150 μl of pre-culture on LB/Chloramphenicol (Chlo – Sigma Aldrich)/Kanamycin (Kan - Sigma Aldrich) agar plates which were further incubated 12 hr. On day 2, a single colony was harvested to inoculate 50 ml of LB/Chlo/Kan for 12 hr at 37°C. On day 3, the 50 ml saturated pre-culture then inoculated 3*1L of Terrific Broth/Chlo/Kan (TB - Formedium) expression culture (using a 2 ml inoculum) which was incubated at 37°C. Upon reaching an OD600 of 0.5–0.8, cultures were ice cooled to 20°C for 10 min then induced with 500 μM of IPTG (Euromedex) for 12 hr after which cultures were centrifuged and stored as dry pellets at −80°C.

## Protein purification

Culture pellets were resuspended in 50 mM Tris pH: 7.5, 300 mM NaCl and 5 μM β-mercaptoethanol (BME) with the addition of complete protease inhibitor (1 tab per 50 mL of lysis buffer). Following resuspension, lysis was performed on ice by sonication for 10 min (30 s on/ 30 s off, 45° amplitude). Clarification was then performed by centrifugation 1 hr at 12,000 g/4°C after which the supernatant was supplemented with 20 mM imidazole and further incubated with 5 ml Ni-NTA resin with a stirring magnet at 4°C for 30 min. Resin retention was performed by gravity with a Bio-Rad glass column after which the resin was washed with 100 mL of washing buffer (50 mM Tris pH: 7.5, 1M NaCl, 2 mM BME and 20 mM Imidazole). His-tagged TgCPSF4-YTH was then eluted by in 50 mM Tris pH: 7.5, 300 mM NaCl, 300 mM Imidazole, 2% glycerol and 2 mM BME and dialyzed overnight with TEV

protease to remove the 8*histidine tag in a buffer composed of 50 mM Tris pH 7.5, 150 mM NaCl, 2% glycerol and 2 mM BME. Non-cleaved forms of TgCPSF4-YTH were removed by flowing through 1 mL of pre-equilibrated Ni-NTA. All subsequent liquid chromatography steps were performed on an Akta Purifier. Nucleic acid contaminants were removed by directly binding CPSF4-YTH onto a 5 ml heparin (GE healthcare) and eluting by a 40 ml step NaCl gradient (150 mM to 2M) in a 50 mM tris pH 7.5, 2 mM BME, 2% glycerol buffer system. Following elution from heparin, fractions of interest were pooled and concentrated to 1 mL using 10 kDa cut-off concentrators before being subsequently injected on an S75 column for size fractionation in a buffer containing 50 mM Tris pH: 7.5, 150 mM NaCl, 1 mM BME. Following the size exclusion step, final fractions were pooled, concentrated to a minimum of 15 mg/mL with centricon 10 kDa concentrators, flash frozen in liquid nitrogen and stored at −80°C.

## Isothermal titration calorimetry

The affinity of at-YTH or tg-YTH for RNA or RNAm6A were determined using a MicroCal iTC200 system (Malvern Panalytical, Malvern, UK). Experiments were performed at 25°C in 20 mM Tris pH 8, 50 mM NaCl. The YTH proteins solutions (at concentration ranging from 25 μM to 60 μM) were loaded in the calorimetric cell. RNA (at concentrations ranging between 0.6 mM to 0.9 mM) were titrated in the protein sample typically by performing 16 injections of 2.5 μL aliquots. Control titrations were performed by titrating RNA into buffer. The dissociation constants (Kd), enthalpy of binding (ΔH), and stoichiometries (N) were obtained after fitting the integrated and normalized data to a single-site binding model. Data were processed using Origin 7.0 (Malvern Panalytical, Malvern, UK). All experiments were performed at least in duplicate to check for reproducibility of the data.

## Crystallization and diffraction

Initial screening was conducted at room temperature using a sitting drop vapor diffusion method combined with automated laser photoablation and direct cryo-cooling (*Zander et al., 2016*) developed by the EMBL Grenoble outstation (CrystalDirect, HTX lab). Apo crystals grew in 1–2 weeks in 20% PEG 4000, 20% isopropanol and 0.1M tri-sodium citrate pH 5.6. The m6A-modified RNA (5 mM, IDT) CPSF4-YTH co-complexes grew in 1 week in 0.1M sodium acetate pH: 4.6 in 40% PEG 200 or 30% PEG 300/400. m6A (5 mM from a 50 mM DMSO solution, Abcam) / CPSF4-YTH co-crystals were grown in 1 week in 50% PEG 400, 0.1M Sodium acetate pH 4.5 and 0.2M LiSO4. All conditions produced high quality diffracting crystals using the CrystalDirect method. M6-adenosine / CPSF4-YTH co crystals were reproduced by hand hanging drop geometry in 24-well VDX plates (Hampton Research) containing 500 ml of mother liquor in each well. Crystal were subsequently fished out in cryo-loops (Hampton) and directly flash-frozen in their crystallization mother liquor. All PEG and associated chemical formulations were obtained from Sigma and purchased as chemical compounds. X-ray diffraction data for m6A / CPSF4-YTH crystals were collected by the autonomous European Synchrotron Radiation Facility (ESRF) beamline MASSIF-1 (*Bowler et al., 2015*; *Svensson et al., 2015*) using automatic protocols for the location and optimal centering of crystals (*Svensson et al., 2018*). Strategy calculations accounted for flux and crystal volume in the parameter prediction for complete datasets. Apo CPSF4-YTH and RNA co-crystals were remotely mounted at the diamond light source beamline IO4. In all cases, diffraction was performed at 100K. Data reduction was performed using XDS (*Kabsch, 2010*) while amplitude scaling/merging was performed by multiple software packages (Aimless or Truncate) by Autoproc or Staraniso (Global Phasing Ltd). Molecular replacement solutions were obtained with Phaser (*McCoy et al., 2007*) (within Phenix) using the crystal structure of *YTHDC1* [Protein Data Bank (PDB) code: 4r3i] as a template. The initial solution was then improved through cycles of manual adjustment in Coot (*Emsley and Cowtan, 2004*), automated building in phenix autobuild (*Terwilliger et al., 2008*) and refined using Refmac5, phenix resolve or Buster (Global Phasing Ltd). Final pdb model corrections were performed using the pdb-redo server (*Joosten et al., 2014*).

## Plant methods

*A. thaliana* Col-0 as well as *cpsf30*-3 (*Pontier et al., 2019*) mutants were cultivated in vitro on plates containing 2.20 g/l synthetic Murashige and Skoog (MS) (Duchefa) medium, pH 5.7, and 8 g/l agar. *fip37-4 LEC1*:FIP37 (*Shen et al., 2016*) were grown on plates containing 4.41 g/l MS medium, pH

5.7, 1% sucrose, and 8 g/l agar; 9-day-old seedlings were collected under a microscope. The seeds were surface-sterilized, sown on plates, incubated for 48 hr at 6°C in the dark, and placed in a growth cabinet at 20°C with a 16-h-d/8-h-dark cycle and 130 µE m$^{-2}$ s$^{-1}$ light (LEDs with white 4500K spectrum, purchased from Vegeled) for 9 days. Total RNA was isolated from 50 mg of seedlings using the TRI reagent (Cat. No. TR-118; Euromedex) according to the manufacturer's instructions.

## Total RNA preparations

Total RNAs were extracted and purified using TRIzol (Invitrogen, Carlsbad, CA, USA) and RNeasy Plus Mini Kit (Qiagen). RNA quantity and quality were measured by NanoDrop 2000 (Thermo Scientific). RNA integrity was assessed by standard non-denaturing 1.2% TBE agarose gel electrophoresis. The mRNA extraction was done using the Dynabeads mRNA purification kit (Thermofisher ref n° 61006) and starting with 80–100 µg of total pooled extracted RNA (from two separated extractions per condition) and following the manufacturers guidelines. Final mRNA yield measurements were done the Qubit HS RNA kit (Thermofisher ref Q32852).

## RNA-seq and sequence alignment

RNA-sequencing was performed in duplicate for each experimental condition, following standard Illumina protocols, by GENEWIZ (South Plainfield, NJ, USA). Briefly, RNA quantity and integrity were determined using the Qubit Fluorometer and the Fragment Analyzer system with the PROSize 3.0 software (Agilent Technologies, Palo Alto, California, USA). The RQN were ranging from 8.8 to 10 for all samples, which was considered sufficient. Illumina TruSEQ RNA library prep and sequencing reagents were used following the manufacturer's recommendations using polyA-selected transcripts (Illumina, San Diego, CA, USA). The samples were sequenced on the Illumina NovaSeq platform (2 x 150 bp, single index) and generated ~20 million paired-end reads for each sample (Table SX). The quality of the raw sequencing reads was assessed using FastQC (http://www.bioinformatics.babra-ham.ac.uk/projects/fastqc/) and MultiQC (*Ewels et al., 2016*). The RNA-Seq reads (FASTQ) were processed and analyzed using the Lasergene Genomics Suite version 15 (DNASTAR, Madison, WI, USA) using default parameters. The paired-end reads were uploaded onto the SeqMan NGen (version 15, DNASTAR. Madison, WI, USA) platform for reference-based assembly using the Toxoplasma Type II ME49 strain (ToxoDB-46, ME49 genome) as reference template. The ArrayStar module (version 15, DNASTAR. Madison, WI, USA) was used for normalization, and statistical analysis of uniquely mapped paired-end reads using the default parameters. The expression data quantification and normalization were calculated using the RPKM (Reads Per Kilobase of transcript per Million mapped reads) normalization method. In parallel, the FASTQ reads were aligned to the ToxoDB-48 build of the *Toxoplasma gondii* ME49 genome (ToxoDB-48) using Subread version 2.0.1 (*Liao et al., 2013*) with the following options 'subread-align -d 50 -D 600 –sortReadsByCoordinates'. Read counts for each gene were calculated using featureCounts from the Subread package (*Liao et al., 2014*). Differential expression analysis was conducted using DESeq2 and default settings within the iDEP.92 web interface (*Ge et al., 2018*). Transcripts were quantified and normalized using TPMCalculator (*Vera Alvarez et al., 2019*).

## m6A quantification by ELISA

Total RNA was isolated from purified METTL3 KD and CPSF30 KD parasites that were treated with vehicle or IAA and the amount of m6A modification was assessed from total RNA using the colorimetric EpiQuik m6A methylation quantification kit as per manufacturer's instructions (EpiGentek, P-9005–96).

## Direct RNA sequencing by nanopore

Nanopore library followed the SQK-RNA002 kit (Oxford Nanopore) recommended protocol, the only modification was the input mRNA quantity increased from 500 to 1000 ng, all other consumables and parameters were standard. Final yields were evaluated using the Qubit HS dsDNA kit (Thermofisher Q32851) with minimum RNA preps reaching at least 150 ng. For all conditions, sequencing was performed on FLO-MIN106 flow cells either using a MinION MK1C or MinION sequencer running Minknow v20.06.5 and guppy v4.09. Basecalling was performed during the run

using the fast-basecalling algorithm with a Q score cutoff >7. Long read alignment (to ME49-Tox-odb-13 and TAIR10 reference fasta files) was performed using Minimap2 (ver 2.1) with the following parameters: '-ax splice -k14 -uf -G 5000 t 10 –secondary=no –sam-hit-only' for *Toxoplasma* and '-ax splice -k14 -uf -G 20000 t 10 –secondary=no –sam-hit-only' for *Arabidopsis*. Aligned reads were converted to bam, sorted, and indexed using Samtools. For *T. gondii* datasets, most sequencing runs were stopped after having generated between 400 k and 500 k of aligned reads to keep a standard of comparison (*T. gondii* reads varying between 30% and 70% of total mRNA depending on the preparation).

## Chimeric transcript detection

Chimeric reads covering two predicted transcripts were extracted using the ChimerID scripts (*Parker et al., 2020*) (https://github.com/bartongroup/Simpson_Barton_Nanopore_1, *Farhat, 2021a* copy archived at swh:1:rev:1b509454a9e25a8c81be5092f8e525ca00e7b5a5) from native RNA Nanopore aligned reads within a Anaconda environment containing the following packages (snakemake ;pysam ; samtools ; scipy ; statsmodels ; py-bgzip ; tabix ; pytables ; bedtools ; minimap2 ; panda=0.25.3). Prior to running the analysis, the ToxoDB gene file (in gff format) was processed by AGAT (https://github.com/NBISweden/AGAT, *Farhat, 2021b* copy archived at swh:1:rev:692791aa30d253bd2ff83397d0dc1a74b5b52adb) to convert to GTF and modify the format. For practical reasons, all KEXXXX contigs were removed from the GTF and genome fasta file. Once extracted, chimeric reads were counted against transcripts in the TGME49 gtf file using Htseq-count. This count file was also supplemented with normal counts of standard transcripts. Data was then normalized using a total aligned read normalization and ratios of Chimera over Total reads were calculated for every transcript in every condition. Log2Fold change in these ratios were then calculated to compare UT against IAA conditions for every KD condition (CPSF4, METTL3, and MORC). Transcripts with readthrough counts below three were excluded from further analysis. Of note, we had no mAID UT condition for MORC so we used the METTL3 UT condition as a comparison to calculate Log2Fch.

## Differential splicing analysis

Splice correction, collapse, quantification and differential isoform representation was performed using the FLAIR pipeline (*Tang et al., 2020*) with standard parameters however keeping non-consistent isoforms after the correction stage. The GTF generated by FLAIR collapse was used to visualize long chimeric transcript formation in Integrated Genome Browser.

## Differential error detection

Differential error detection was performed using the Epinano pipeline 1.2 (available code at, described in *Liu et al., 2019*). To limit unwanted background errors, DRS fast5 raw datafiles (of METTL3 UT and METTL3 24h-IAA or Arabidopsis WT and FIP37 ko) were re-basecalled on the MinionMK1C using the high accuracy basecalling option. Bam alignment files were generated with the following command "minimap2 -MD -ax splice -k14 -uf -t 12 ref-genome.fasta Guppy-HA-reads.fastq | samtools view -hbS -F 3844 - | samtools sort -@ 6 -o Alignement.bam'. Briefly, nucleotide variants were calculated on the aligned bam file for UT and KD/KO using the Epinano_Variants.py python script with normal parameters. A global error statistic by nucleotide position was calculated using the Epinano_sumErr.py python script with a minimum of 50 reads per position. Finally, differential error z-scores were obtained by comparing the two datasets with the Rscript Epinano_DiffErr.R, the generated csv file was trimmed of unmodified residues using the linux « grep » command to select only the « mod » containing lines which have a z-score above 3. For representation, the generated data was converted to a bedgraph format representing nucleotide positions as a function of the epinano_DiffErr z-score * 10.

## Motif detection at differential error peaks

The most significant peaks were selected based on a z-score higher than 8, from the initial position, 10 nucleotides upstream and downstream were selected to generate a FASTA file using 'bedtools getfasta'. For negative sense transcripts, the sequences were reverse complemented using seqkit. Duplicates were removed using seqkit rmdup and further curated manually to avoid duplications.

This created a FASTA file containing 223 sequences which were summitted as stranded to the MEME server (https://meme-suite.org/meme/) which detected only one significant motif RRACD present within 141 sequences with a E-value of 7.4e-25. The same approach was used on *Arabidopsis* WT / Fip37L-KD on a total of 883 unique peaks and generated a similar motif with E-value of 7.4e-186.

## Acknowledgements

The HTX Lab (EMBL Grenoble) are thanked for support in screening for crystal conditions and automatic mounting of crystals. We thank Diamond Light Source for beamtime and David Aragao for support. This work used the platforms of the Grenoble Instruct-ERIC center (ISBG; UAR 3518 CNRS-CEA-UGA-EMBL) within the Grenoble Partnership for Structural Biology (PSB), supported by FRISBI (ANR-10-INBS-0005–02) and GRAL, financed within the University Grenoble Alpes graduate school (Ecoles Universitaires de Recherche) CBH-EUR-GS (ANR-17-EURE-0003).

## Additional information

### Funding

| Funder | Grant reference number | Author |
|---|---|---|
| Agence Nationale de la Recherche | ANR-11-LABX-0024 | Dayana C Farhat<br>Mohamed-Ali Hakimi<br>Christopher Swale |
| Agence Nationale de la Recherche | ANR-18-CE15-0023 | Charlotte Corrao<br>Alexandre Bougdour<br>Mohamed-Ali Hakimi<br>Christopher Swale |
| European Research Council | ERC Consolidator Grant N° 614880 Hosting TOXO | Mohamed-Ali Hakimi |
| Fondation pour la Recherche Médicale | FRM FDT201904008364 | Dayana C Farhat<br>Mohamed-Ali Hakimi |
| Agence Nationale de la Recherche | ANR-10-INBS-08 | Lucid Belmudes<br>Yohann Couté |
| Agence Nationale de la Recherche | ANR-10-LABX-41 | Dominique Pontier<br>Thierry Lagrange |
| Agence Nationale de la Recherche | ANR-18-EURE-0019 | Dominique Pontier<br>Thierry Lagrange |

The funders had no role in study design, data collection and interpretation, or the decision to submit the work for publication.

### Author contributions

Dayana C Farhat, Formal analysis, Validation, Investigation, Visualization, Methodology, Writing - original draft; Matthew W Bowler, Conceptualization, Data curation, Formal analysis, Funding acquisition, Visualization, Writing - review and editing; Guillaume Communie, Data curation, Software, Formal analysis; Dominique Pontier, Investigation, Visualization, Methodology, Writing - review and editing; Lucid Belmudes, Caroline Mas, Data curation, Formal analysis, Investigation; Charlotte Corrao, Formal analysis, Investigation, Methodology; Yohann Couté, Conceptualization, Formal analysis, Funding acquisition, Investigation, Methodology; Alexandre Bougdour, Formal analysis, Investigation, Visualization, Methodology, Writing - review and editing; Thierry Lagrange, Conceptualization, Formal analysis, Supervision, Funding acquisition, Writing - original draft, Project administration, Writing - review and editing; Mohamed-Ali Hakimi, Conceptualization, Formal analysis, Supervision, Funding acquisition, Validation, Investigation, Visualization, Methodology, Writing - original draft, Project administration, Writing - review and editing; Christopher Swale, Conceptualization, Data curation, Software, Formal analysis, Supervision, Validation, Investigation, Visualization, Methodology, Writing - original draft, Project administration, Writing - review and editing

## Author ORCIDs
Matthew W Bowler ![ORCID] http://orcid.org/0000-0003-0465-3351
Yohann Couté ![ORCID] http://orcid.org/0000-0003-3896-6196
Alexandre Bougdour ![ORCID] http://orcid.org/0000-0002-5895-0020
Mohamed-Ali Hakimi ![ORCID] https://orcid.org/0000-0002-2547-8233
Christopher Swale ![ORCID] https://orcid.org/0000-0002-9739-7774

## Decision letter and Author response
Decision letter https://doi.org/10.7554/eLife.68312.sa1
Author response https://doi.org/10.7554/eLife.68312.sa2

# Additional files

## Supplementary files
• Supplementary file 1. Table: Mass spectrometry quantitation of proteins from *T. gondii* CPSF and m6A writer complexes. Mass spectrometry-based proteomic analysis of Flag elution identified CPSF1, WDR33, Fip1, PAP, METTL3, METTL14, and WTAP and their partners. The identities of the proteins (accession number on ToxoDB, gene name, and description) are indicated. For each protein, MaxQuant reports an intensity-based absolute quantification (iBAQ) value, a measure of protein abundance.

• Supplementary file 2. Table: Crystallography data statistics.

• Supplementary file 3. Table: Analysis of Illumina high-throughput RNA-Seq data of the CPSF4 KD parasite line of *T. gondii*, related to *Figure 7—figure supplement 1B*. RNA-Seq report, Raw counts of *T. gondii* transcripts generated by feature Counts (Subread), Normalized *T. gondii* transcripts (TPM), Differential expression analysis (iDEP.92), data used in volcano plot (48 hr vs UT).

• Supplementary file 4. Figure – Read-through events and chimeric RNAs detected on *T. gondii* chromosomes following CPSF4 and METTL3 post-translational knock-down.

• Supplementary file 5. Figure - Read-through events and chimeric RNAs detected on *A. thaliana* chromosome 1, 2, and 5 in the context of *fip37L* and *cpsf30-3* mutants.

• Supplementary file 6. Table: *T. gondii* strains, vectors, and primers. List of *T. gondii* parasite strains and transgenic lines as well as plasmids used in this work. Primers and DNA construct used in this work are also charted in the table.

• Supplementary file 7. Full wwPDB X-ray structure validation report of crystal structure of *Toxoplasma* CPSF4-YTH domain in apo form (PDB ID: 7NG2).

• Supplementary file 8. Full wwPDB X-ray structure validation report of crystal structure of *Toxoplasma* CPSF4-YTH domain bound to m6A (PDB ID: 7NH2).

• Supplementary file 9. Full wwPDB X-ray structure validation report of crystal structure of the Toxoplasma CPSF4 YTH-domain in complex with a seven mer m6A-modified RNA (PDB ID: 7NJC).

• Transparent reporting form

## Data availability
The Nanopore RNAseq data have been deposited in NCBI's SRA data PRJNA705300. The MS proteomics data have been deposited to the ProteomeXchange Consortium through the PRIDE partner repository with the dataset identifier PXD024326. Sequencing data have been deposited in GEO under accession code GSE168155.

The following datasets were generated:

| Author(s) | Year | Dataset title | Dataset URL | Database and Identifier |
|---|---|---|---|---|
| Hakimi | 2021 | Raw reads of direct RNA sequencing of Toxoplasma | https://www.ncbi.nlm.nih.gov/bioproject/? | NCBI BioProject, PRJNA705300 |

| | | gondii and *Arabidopsis thaliana* in the context m6A writter/reader protein knockouts | term=PRJNA705300 | |
| --- | --- | --- | --- | --- |
| Hakimi | 2021 | Interactome of Toxoplasma gondii Pre-mRNA 3' Processing Complexes | https://www.ebi.ac.uk/pride/archive/projects/PXD024326 | PRIDE, PXD024326 |
| Hakimi | 2021 | A plant-like mechanism coupling m6A reading to polyadenylation safeguards transcriptome integrity and developmental genes partitioning in Toxoplasma | https://www.ncbi.nlm.nih.gov/geo/query/acc.cgi?acc=GSE168155 | NCBI Gene Expression Omnibus, GSE168155 |

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
