## [Decision Letter]

**Acceptance summary:**

This paper describes a novel regulatory mechanism coupling RNA m6A modification to polyadenylation site selection and transcriptional termination, which is shared between apicomplexan parasites (*Toxoplasma gondii*) and plants (*Arabidopsis thaliana*). The data are supported by multi-angle approaches (biochemistry, structural biology, functional genetics and genomics) and are of wide interest to audience studying gene and RNA regulation, but also communities who develop parasite-targeting strategies. These findings highlight how transcription barriers are maintained upon intense replication of a parasitic gene-dense genome and may explain how these organisms quickly adapt to external stimuli.

**Decision letter after peer review:**

Thank you for submitting your article "A plant-like mechanism coupling m6A reading to polyadenylation safeguards transcriptome integrity and developmental genes partitioning in Toxoplasma" for consideration by *eLife*. Your article has been reviewed by 3 peer reviewers, and the evaluation has been overseen by a Reviewing Editor and Dominique Soldati-Favre as the Senior Editor. The reviewers have opted to remain anonymous.

Essential Revisions:

1) Regarding the connection between m6A RNA methylation and transcriptional readthrough: please provide accurate quantifications, as listed by reviewer #1 and comment 12 from reviewer#3.

2) Quantifications are also needed in the knock down experiments: please provide quantitative WB rather than IF. Also refer to reviewer#3 comments to try to improve detection of m6A changes, beyond poorly quantifiable IF data (m6A ELISA could also be attempted?).

3) Similarly, quantitative measures should be provided to describe the phenotypic overlap between the METTL3 KD and CPSF4 KD (Comment 13 reviewer #3) and the pulldown overlap between the different protein complexes (comment 14 reviewer #3).

4) Do you have evidence to support that m6A is developmentally regulated in *T. gondii*?

5) Reviewers raised similar comments about the writing: sentences are often too lengthy, too internally convoluted and deploy sesquipedalian vocabulary, rendering their meaning unclear and the message the authors want to convey difficult to understand. The text should be simplified and streamlined wherever possible.

6) Provide explanations to specific questions raised by the different reviewers.

*Reviewer #1:*

In this rich and multidimensional manuscript, Farhat et al. report a role for m6A in controlling transcriptional termination in *T. gondii*. The authors combine biochemistry, structural biology, genetic and genomic approaches to characterize CPSF4, a protein forming part of the cleavage and polyadenylation complex in *T. gondii*. The authors demonstrate that using its YTH binding domain it selectively binds m6A-harboring targets. Using genetic disruption of this protein or m6A-machinery components, the authors demonstrate that loss of m6A leads to widespread levels of transcriptional readthrough.

In general, we very much enjoyed this manuscript. It establishes a novel mode of regulation mediated by m6A, and demonstrates how such a mode of regulation (which is not present in human) can evolve, namely by the fusion of an m6A binding domain to an RNA binding protein.

Key comments refer to the genomic aspects of the paper. In this section the authors seek to ascribe a causal role for m6A in controlling transcriptional termination. This section is rich in qualitative descriptions and accompanying figure panels (figures 6 to 10 and many supplementaries are nearly entirely based on screenshots). However, it lacks quantitations of the signal and statistics that are critical in order to evaluate the weight of the evidence in favor of the proposed model. Specifically, is remains unclear :

– In how many genes is transcriptional readthrough observed, in each of the three mutants?

– What is the extent of overlap between the three? Is it statistically significant, taking into account the fact that the nanopore based approach is heavily dependent on expression levels?

– How many m6A sites are identified by the authors? How are they distributed within genes? Other than the reported enrichment in 3' UTR, are they also enriched near the stop codons as has been reported in mammals?

– What fraction of the m6A sites are in genes with evidence for transcriptional readthrough? and in genes without evidence for transcriptional read-through? Is the overlap between m6A and readthrough statistically significant?

– Are genes that are methylated at higher levels also subject to more readthrough upon loss of the methylation machinery?

– At the single molecule level (via nanopore): can the authors distinguish between the methylated and unmethylated molecules, and assess whether readthrough is only observed in the latter molecules?

In our view it is critical to quantitatively address these questions, to provide the extent of support for a direct relationship between m6A and termination.

*Reviewer #2:*

Although the function of m6A in mRNA stability and translation has been well established, the implication of m6A in the control of polyadenylation site selection remains obscure. Dayana C. Farhat et al. first identified the components of the core CPSF complex within *T. gondii*, an obligate parasite who can turn into major thread to the unborn and to immunocompromised people. Then they used biochemistry and structural methods to prove that CPSF4 was an m6A reader, coupling m6A modification directly to APA events. Finally, they detected transcriptional readthrough events upon CPSF4 or Mettl3 KD using Nanopore DRS. Most importantly, they detected m6A-dependent PAS was an efficient transcriptional barrier, that preventing aberrant readthrough of highly expressed genes into the downstream repressed ones in the high gene density genome. And this will make the parasites to reproduce more quickly.

The conclusions of this paper are mostly well support by the data, and this study opens up new insights about this ancient APA regulation. It helps us to understand how parasites and plants adjust themselves quickly under external stimuli.

The conclusions of this paper are mostly well support by the data, but some aspects need to be clarified.

1. What's the relationship between ZnF recognition and YTH recognition of m6A within CPSF4? Because ZnF of CPSF4 in human is found to recognize AAUAA site directly, but AAUAA site was also reported to be m6A modified, which is recognized by YTH domain of CPSF4.Do you have evidence to show PAS binding by these two domains within one protein are competing or assisting each other?

2. Knocking down of CPSF4 or Mettl3 would have some indirect side effects since these two proteins might involve in other biological processes, I'm wondering if you could test the 3'-end profiling using CPSF4-m6A-binding-defective mutants.

3. In the paper cited by the authors (doi: 10.7554/*eLife*.49658), nanopore DSR was also used to map the mRNA profiling in the m6A writer(vir-1) defective background in Arabidopsis, but opposite conclusions were reached: could the authors comment on this inconsistency?

4. As you compared the structure of TgYTH-m6A RNA with other complexes, and identified RNA bound within a clearly charged groove as seen in other structures. What's the point of a potential secondary binding groove as stated in the text? What do you mean by referring to "multiple binding modes" in line724?

*Reviewer #3:*

In this paper, Farhat et al. demonstrate that m6A on mRNA in *Toxoplasma gondii* interacts with a cleavage and polyadenylation factor protein (CPSF4). This interaction is mediated via the YTH-domain and is characterized using high resolution crystal structures and isothermal calorimetry. The authors have identified the CPSF4 complex using reciprocal IP-MS. The also carried out an IP-MS for the m6A writer machinery – namely METTL3, METTL14, and WTAP homologs to reflect the m6A writer machinery. The main findings for this paper relate to the fact that a loss of or CPSF4 YTH reader protein results in the production of chimeric mRNAs as a result of aberrant transcriptional termination in *T. gondii* and *Arabidopsis thaliana*. This phenomenon is also phenocopied in the loss of METTL3 writer protein. Of interest, transcripts which are reliant upon this m6A-dependent recruitment of CPSF4 are generally upstream of otherwise repressed developmental stage-governed transcripts.

While the paper presents a large amount of data, some of excellent quality, there are critical issues that need to be addressed. Of importance, the implication that METTL3 or CPSF4 KD is involved in developmental regulation requires a clear demonstration of a phenotype where this is impaired and that m6A site is dynamically regulated at the boundary of these transcripts. This involves demonstrating stage transition is impaired in the knockdowns and also using nanopore or other technologies demonstrate that the m6A site is "dynamic" despite not possessing known demethylases.

The effort to combine multiple strands of work ranging from polyadenylation and transcript termination, the role of m6A in this process and alternative splicing in aberrant transcripts and finally development made the story confusing and often detracted from the important message that could be conveyed. Also, it meant that some conclusions were not well supported by the data.

Here is a list of some of the essential issues to address:

1. It is essential to show quantifiable western blot data that shows the downregulation of the different target proteins in the knock down cell lines.

2. m6A quantification cannot be carried out using IFAs. If using antibodies – show specificity as well. M6A occurs in the host cell as well – why is this not reflected in the staining Figure 3A? I would suggest to either selectively enrich for TG mRNA and then carry out dot blots or mass spectrometry or perhaps even sequencing based approaches. This is critical since none of the IFA data presented is quantifiable.

3. The link between developmental regulation and m6A is poorly supported. As a minimum it would be necessary to show that m6A levels are developmentally regulated.

4. Also, there is a significant challenge on how the knock down data can be interpreted. The m6A knockdown would have a global impact of this modification in all mRNA locations. So it is not clear on whether the impact observed is a direct or indirect result.

5. "Of the CPSF complex, the *T. gondii* CPSF4 subunit can be distinguished as one holding a unique architecture which interestingly is shared with the plant CPSF4 family, and it constitutes of a co-occurrence of three zinc fingers and a conserved YTH domain" – Which YTH domain? DC or DF?

6. In this context I could imagine that the replacement of CPSF4 with a version lacking the YTH domain may be a way to clearly demonstrate the importance of the YTH domain.

7. Binding affinities using ITC – TG CPSF4 with m6A has a Kd at 5uM for the modified oligo and Arabidopsis CPSF4 shows close to a Kd 6uM for the non-m6A modified oligo. Could the authors explain this judgement on what constitutes good/bad binding?

8. In addition, the specificity of YTH binding to m6A is based on a single oligonucleotide sequence. At least a scrambled version of this should be used.

9. Based on the crystallization results where CPSF4 does not show RNA context specificity, except for m6A residue, how do the authors predict that this recruitment IS context specific? There exist other m6A sites in regions that do not constitute the 3' UTR (Line 612).

10. Figure 7A -An IFA is not sufficient. A western blot is needed in the very least to validate the KD.

11. Figure 7B – Are these adjusted p-values. Then these would not be a relatively low number of differentially expressed genes then as per Line 457.

12. Line 480 – There must be a way to quantify these readthrough events. A statement like this requires substantiation in the form of data.

13. Please provide a quantitative methodology to depict how METTL3 KD phenocopies CPSF4 KD. How many transcripts display aberrant splicing and which of these are common to both KDs?

14. Line 209 – Could you draw a venn diagram or similar representation to see the overlapping proteins identified in the complex in each pulldown?

15. Since you identified Val522 as an additional amino acid for the binding cage of m6A when compared to its counterparts, I suggest you to test the binding affinity towards m6A-modified RNA using V522 mutation by ITC.

---

## [Author Response]

Essential Revisions:1) Regarding the connection between m6A RNA methylation and transcriptional readthrough: please provide accurate quantifications, as listed by reviewer #1 and comment 12 from reviewer #3.

We have now added a more descriptive quantification of readthrough events in Figure 8 — figure supplement 1 of the manuscript. The full response to these comments can be found in the response to reviewer 1 below.

2) Quantifications are also needed in the knock down experiments: please provide quantitative WB rather than IF. Also refer to reviewer #3 comments to try to improve detection of m6A changes, beyond poorly quantifiable IF data (m6A ELISA could also be attempted?).

To answer this request, we added western blot quantifications showing the disappearance over time of METTL3 (new Figure 3B) and CPSF4 (new Figure 7B) proteins after auxin addition.

– We agree that immunofluorescence is not quantitative. The dot blots in our hands give poor results. We opted for the ELISA technique based on the EpiQuick kit developed by EpiGentek. The challenge for us was to properly purify the tachyzoites (by sequential filtrations) to subtract the host cell background. The data reported in the new Figure 3D show a significant decrease in m6A in total Toxoplasma RNAs only after METTL3 depletion, while depletion of CPSF4 does not alter m6A levels.

3) Similarly, quantitative measures should be provided to describe the phenotypic overlap between the METTL3 KD and CPSF4 KD (Comment 13 reviewer #3) and the pulldown overlap between the different protein complexes (comment 14 reviewer #3).

The phenotypic overlap is now illustrated by the Venn diagram showing specific and common chimera enrichment in Figure 8C. A more detailed response can also be found in the response to reviewer 1.

– A Venn diagram (new Figure 1C) now details overlapping proteins identified in the complex in each pulldown as requested by reviewer 3.

4) Do you have evidence to support that m6A is developmentally regulated in T. gondii?

Reviewer 3's comment is very relevant since developmental trajectories and stage transitions are accompanied by major transcriptome changes in Toxoplasma gondii. Yet, depletion of METTL3 or CPSF30 does not promote differentiation to chronic or sexual stages, unlike MORC KD we used as a control. However, CPSF30 or METTL3 KD in the tachyzoite reveals an unexpected role for m6Adependent 3’ end polyadenylation that likely serves as a transcriptional barrier in tachyzoite at closely and tandemly-arranged stage specific genes. Indeed, a significant number of readthroughs caused by depletions of CPSF30 or METTL3 originate from the expression of tachyzoite gene and invade adjacent gene whose expression is restricted to chronic or sexual stages. Therefore, we did not claim on a role for m6A in stage transitions but we did highlight a function for the modification in developmental gene partitioning at least in tachyzoite.

Whether this partition remains in the other stages is a burning question, still to be answered. According to Holmes et al. (1), no difference in m6A levels was found when comparing tachyzoite to early bradyzoite (converted in vitro). One of the challenges to overcome is to be able to modulate the writing/reading of m6A in chronic or sexual stages, which have long been inaccessible without recourse to animal models. With the discovery of BFD1 (2) and MORC (3), culture models mimicking the mouse brain or cat gut offer new opportunities to explore m6A at other stages of development than tachyzoite.

1 – Holmes et al., – doi: https://doi.org/10.1101/2021.01.29.428772

2 – Wadman et al., Cell, 2020

3 – Farhat et al., Nature Microbiology, 2020

5) Reviewers raised similar comments about the writing: sentences are often too lengthy, too internally convoluted and deploy sesquipedalian vocabulary, rendering their meaning unclear and the message the authors want to convey difficult to understand. The text should be simplified and streamlined wherever possible.

The text has been edited by native English speakers and simplified as requested.

6) Provide explanations to specific questions raised by the different reviewers.

We have responded to each reviewer's comment with the utmost care.

Reviewer #1:In this rich and multidimensional manuscript, Farhat et al. report a role for m6A in controlling transcriptional termination in T. gondii. The authors combine biochemistry, structural biology, genetic and genomic approaches to characterize CPSF4, a protein forming part of the cleavage and polyadenylation complex in T. gondii. The authors demonstrate that using its YTH binding domain it selectively binds m6A-harboring targets. Using genetic disruption of this protein or m6A-machinery components, the authors demonstrate that loss of m6A leads to widespread levels of transcriptional readthrough.In general, we very much enjoyed this manuscript. It establishes a novel mode of regulation mediated by m6A, and demonstrates how such a mode of regulation (which is not present in human) can evolve, namely by the fusion of an m6A binding domain to an RNA binding protein.Key comments refer to the genomic aspects of the paper. In this section the authors seek to ascribe a causal role for m6A in controlling transcriptional termination. This section is rich in qualitative descriptions and accompanying figure panels (figures 6 to 10 and many supplementaries are nearly entirely based on screenshots). However, it lacks quantitations of the signal and statistics that are critical in order to evaluate the weight of the evidence in favor of the proposed model. Specifically, is remains unclear:

We acknowledge the lack in precision for the quantification of the readthrough events, to be more precise, we have performed a number of additional analyses and will set out to answer the following points raised. We will also include a more detailed discussion on the limitations of these quantifications. We are indeed probably measuring only a fraction of readthroughs as direct RNA Nanopore sequencing is specific towards fully poly-adenylated mRNA. As such, all the transcriptional readthrough events resulting in non poly-adenylated mRNAs can never be quantified.

– In how many genes is transcriptional readthrough observed, in each of the three mutants?

A Venn diagram (introduced in new Figure 8C) now details the number of genes with significant relative increases in read-throughs in the 3 knockdown cell lines as well as a general distribution of Log2 fold change enrichment. (new Figure 8 – supplement 1B)

– What is the extent of overlap between the three? Is it statistically significant, taking into account the fact that the nanopore based approach is heavily dependent on expression levels?

When data are not filtered, brute correlations in chimera detection correlate clearly between METTL3 and CPSF4 KD with a R2 of 0.887 (new Figure 8B and Figure 8 – supplement 1A). As indeed our ability to detect chimeras is dependent on expression levels, we decided to calculate first relative chimeric ratios (chimeras over total transcripts) then use this ratio to calculate fold changes between UT and IAA conditions. When we refine the data further based on strong Log2 fold change values with a least 3 counts of chimeric reads observed, the total number of genes is less than 200.

As you can see, most of METTL3 read-throughs are common to CPSF4, CPSF4 has a more pronounced readthrough phenotype and an important number of exclusive readthroughs. This is quite logic as knocking down CPSF4 will also result in dysfunctional 3’ end termination in m6A independent PAS.

– How many m6A sites are identified by the authors? How are they distributed within genes? Other than the reported enrichment in 3' UTR, are they also enriched near the stop codons as has been reported in mammals?

More details on differential error distribution are now visible in a new figure 10C. As the differential error score was calculated on at least 50 nanopore reads, only around 1000 genes could be analysed. For the most part, error distribution is observed on transcription termination sites (new Figure 10C).

– What fraction of the m6A sites are in genes with evidence for transcriptional readthrough? and in genes without evidence for transcriptional read-through? Is the overlap between m6A and readthrough statistically significant?

We find no statistical correlation between error site presence and the propensity to form readthroughs. This is due to the fact that only a very small fraction of METTL3- mAID genes producing readthroughs had sufficient coverage for differential error calculation. As you can see from the scaled distribution over transcripts, differential error sites are mostly found at the transcription termination sites.

– Are genes that are methylated at higher levels also subject to more readthrough upon loss of the methylation machinery?

As stated above, there does not seem to be a correlation between higher methylations and readthrough formation despite the fact that we have observed putative error sites just on the beginning of some readthroughs.

– At the single molecule level (via nanopore): can the authors distinguish between the methylated and unmethylated molecules, and assess whether readthrough is only observed in the latter molecules?

Differential error significance per base is produced from at least a 50 read pileup and represents an overall assessment. Due to the inherently high error rate in nanopore reads, it is currently impossible to assess an m6A site on a single molecule.

In our view it is critical to quantitatively address these questions, to provide the extent of support for a direct relationship between m6A and termination.Reviewer #2:Although the function of m6A in mRNA stability and translation has been well established, the implication of m6A in the control of polyadenylation site selection remains obscure. Dayana C. Farhat et al. first identified the components of the core CPSF complex within T. gondii, an obligate parasite who can turn into major thread to the unborn and to immunocompromised people. Then they used biochemistry and structural methods to prove that CPSF4 was an m6A reader, coupling m6A modification directly to APA events. Finally, they detected transcriptional readthrough events upon CPSF4 or Mettl3 KD using Nanopore DRS. Most importantly, they detected m6A-dependent PAS was an efficient transcriptional barrier, that preventing aberrant readthrough of highly expressed genes into the downstream repressed ones in the high gene density genome. And this will make the parasites to reproduce more quickly.The conclusions of this paper are mostly well support by the data, and this study opens up new insights about this ancient APA regulation. It helps us to understand how parasites and plants adjust themselves quickly under external stimuli.The conclusions of this paper are mostly well support by the data, but some aspects need to be clarified.1. What's the relationship between ZnF recognition and YTH recognition of m6A within CPSF4? Because ZnF of CPSF4 in human is found to recognize AAUAA site directly, but AAUAA site was also reported to be m6A modified, which is recognized by YTH domain of CPSF4.Do you have evidence to show PAS binding by these two domains within one protein are competing or assisting each other?

This is a very interesting point, for which there is probably no definitive answer. In higher eukaryotes, a canonical PAS signal is found within approximately 70% of transcripts (Neve et al., 2017). In apicomplexan parasites, no clear PAS signal has been characterized as of yet and we only could see clear m6A sites on the canonical RRAC motif recognized by METTL3/METTL14. In any case, if we reason by homology to the CPSF1/CPSF4/WDR33 structures interacting with the AUAAA motif (Sun et al., 2018; Clerici et al., 2018), all the adenosines involved in this interaction would not be available for binding to the YTH domain if they were m6A modified. As such, you would have a binding competition if the m6A was on the PAS site directly. If the m6A is in close proximity, then a cooperative binding mode can be imagined where the zinc fingers would bind the PAS and the YTH domain would bind a proximal m6A.

2. Knocking down of CPSF4 or Mettl3 would have some indirect side effects since these two proteins might involve in other biological processes, I'm wondering if you could test the 3'-end profiling using CPSF4-m6A-binding-defective mutants.

This concern of pleiotropic effects which we also have is fully understandable, as stated in the response to reviewer 3 point 6 we have tried deleting the YTH module within CPSF4 unsuccessfully. Because on the very week expression of CPSF core subunits combined with their paramount essentiality, it is extremely difficult to generate any altered versions of these proteins and select them within viable parasites.

3. In the paper cited by the authors (doi: 10.7554/eLife.49658), nanopore DSR was also used to map the mRNA profiling in the m6A writer(vir-1) defective background in Arabidopsis, but opposite conclusions were reached: could the authors comment on this inconsistency?

It is not clear what inconsistency the reviewer is referring to, as our data parallel those

of Parker et al., (2020; doi: 10.7554/eLife.49658) although generated in different genetic backgrounds. For instance, we were able to confirm the m6A signature (AGAUU) that was identified by Parker et al., using Nanopore DRS data.

Loss of m6A from 3’ untranslated regions was associated by Parker et al., with decreased relative transcript abundance and defective RNA 3’ end formation. They detected 523 loci with increased levels of chimeric RNAs in vir-1 resulting from unterminated transcription proceeding into downstream genes on the same strand. In the same fashion, we were able to confirm the existence of single full-length chimeric transcripts in fip37L and cpsf30-3 mutants. Both studies found that disrupting m6A was associated with altered patterns of mRNA cleavage and polyadenylation.

The predominant effect in their study was the activation of proximal poly(A) sites in the absence of m6A, indicating that m6A can suppress proximal poly(A) site selection. In our study, impairing m6A writing (fip37L) or reading (cpsf30-3) cause a readthrough of an upstream gene, the transcription of which did not terminate and read into the adjacent gene and terminated at the PAS of this latter instead. Parker have taken the analysis a step further by showing that only 33% of upstream genes forming the chimeric RNAs had detectable m6A sites in the VIRcomplemented line with restored VIR activity. They hypothesize the existence of an m6A-independent role for VIR (or the writer complex) in 3’ end formation or an indirect effect on factors required for 3’ processing.

4. As you compared the structure of TgYTH-m6A RNA with other complexes, and identified RNA bound within a clearly charged groove as seen in other structures. What's the point of a potential secondary binding groove as stated in the text? What do you mean by referring to "multiple binding modes" in line724?

To be clear, the second positively charged groove is clearly visible but its function is only speculated. What is certain is that crystal structures of isolated YTH domains in complex with usually short RNA chains (which is required to generate crystalizing complexes) show clearly the m6A recognition ability but fall short to truly explain interactions with extended RNA chains or in the context of larger complexes such as in our case the CPSF complex. These complexes are quite modular and can possibly adapt to different RNA motifs and secondary structures. As such, alternative binding modes are possible. To further expand on this point, we have now added an additional sentence within the discussion (Line 717).

Reviewer #3:In this paper, Farhat et al. demonstrate that m6A on mRNA in Toxoplasma gondii interacts with a cleavage and polyadenylation factor protein (CPSF4). This interaction is mediated via the YTH-domain and is characterized using high resolution crystal structures and isothermal calorimetry. The authors have identified the CPSF4 complex using reciprocal IP-MS. The also carried out an IP-MS for the m6A writer machinery – namely METTL3, METTL14, and WTAP homologs to reflect the m6A writer machinery. The main findings for this paper relate to the fact that a loss of or CPSF4 YTH reader protein results in the production of chimeric mRNAs as a result of aberrant transcriptional termination in T. gondii and Arabidopsis thaliana. This phenomenon is also phenocopied in the loss of METTL3 writer protein. Of interest, transcripts which are reliant upon this m6A-dependent recruitment of CPSF4 are generally upstream of otherwise repressed developmental stage-governed transcripts.While the paper presents a large amount of data, some of excellent quality, there are critical issues that need to be addressed. Of importance, the implication that METTL3 or CPSF4 KD is involved in developmental regulation requires a clear demonstration of a phenotype where this is impaired and that m6A site is dynamically regulated at the boundary of these transcripts. This involves demonstrating stage transition is impaired in the knockdowns and also using nanopore or other technologies demonstrate that the m6A site is "dynamic" despite not possessing known demethylases.The effort to combine multiple strands of work ranging from polyadenylation and transcript termination, the role of m6A in this process and alternative splicing in aberrant transcripts and finally development made the story confusing and often detracted from the important message that could be conveyed. Also, it meant that some conclusions were not well supported by the data.Here is a list of some of the essential issues to address:1. It is essential to show quantifiable western blot data that shows the downregulation of the different target proteins in the knock down cell lines.

As suggested, we added western blot data for both METTL3 (new Figure 3B) and CPSF30 (new Figure 7B) KD.

2. m6A quantification cannot be carried out using IFAs. If using antibodies – show specificity as well. M6A occurs in the host cell as well – why is this not reflected in the staining Figure 3A? I would suggest to either selectively enrich for TG mRNA and then carry out dot blots or mass spectrometry or perhaps even sequencing based approaches. This is critical since none of the IFA data presented is quantifiable.

We agree that immunofluorescence is not quantitative. The dot blots in our hands give poor results. We then opted for the ELISA technique based on the EpiQuick kit developed by EpiGentek. The challenge for us was to properly purify the tachyzoites (by sequential filtrations) to subtract the host cell background. The data reported in the new Figure 3D show a significant decrease in m6A in total Toxoplasma RNAs exclusively after METTL3 depletion, as the decrease in CPSF4 protein (used as a control) does not alter m6A levels.

3. The link between developmental regulation and m6A is poorly supported. As a minimum it would be necessary to show that m6A levels are developmentally regulated.

Thank you for this thoughtful comment. Indeed, developmental trajectories and stage transitions are accompanied by major transcriptome changes in Toxoplasma gondii. Yet, depletion of METTL3 or CPSF30 does not promote differentiation to chronic or sexual stages, unlike MORC KD we used as a control. However, CPSF30 or METTL3 KD in the tachyzoite reveals an unexpected role for m6Adependent 3’ end polyadenylation that likely serves as a transcriptional barrier in tachyzoite at closely and tandemly-arranged stage specific genes. Indeed, a significant number of readthroughs caused by depletions of CPSF30 or METTL3 originate from the expression of tachyzoite gene and invade adjacent gene whose expression is restricted to chronic or sexual stages. Therefore, we did not claim on a role for m6A in stage transitions but we did highlight a function for the modification in developmental gene partitioning at least in tachyzoite.

Whether this partition remains in the other stages is a burning question, still answered. According to Holmes et al. (1), no difference in m6A levels was found when comparing tachyzoite to early bradyzoite (converted in vitro). One of the challenges to overcome is to be able to modulate the writing/reading of m6A in chronic or sexual stages, which have long been inaccessible without recourse to animal models. With the discovery of BFD1 (2) and MORC (3), culture models mimicking the mouse brain or cat gut offer new opportunities to explore m6A at other stages of development than tachyzoite.

1- Holmes et al., – doi: https://doi.org/10.1101/2021.01.29.428772

2- Wadman et al., Cell, 2020

3- Farhat et al., Nature Microbiology, 2020

4. Also, there is a significant challenge on how the knock down data can be interpreted. The m6A knockdown would have a global impact of this modification in all mRNA locations. So it is not clear on whether the impact observed is a direct or indirect result.

The re-analysis provided as an answer to the reviewer 1 (especially the new Figure 8C with the Venn diagram) should enlighten this question and a non-negligible amount of METTL3 and CPSF4 KD read-throughs are commonly observed.

5. "Of the CPSF complex, the T. gondii CPSF4 subunit can be distinguished as one holding a unique architecture which interestingly is shared with the plant CPSF4 family, and it constitutes of a co-occurrence of three zinc fingers and a conserved YTH domain" – Which YTH domain? DC or DF?

DF and DC YTH are quite closely related but to be precise and as detailed in the answer to reviewer 1, the CPSF4 YTH is closer to a YTH-DC2. This is quite logic as it remains a nuclear YTH domain although associated with the CPSF complex.

6. In this context I could imagine that the replacement of CPSF4 with a version lacking the YTH domain may be a way to clearly demonstrate the importance of the YTH domain.

This is a valid observation, as the YTH domain is in the C-terminus we tried on several occasions to insert a HA/FLAG/DHFR/stop prior to the YTH domain but were unable to select any positive clones. We assume that this impacts the fitness of parasites too much and is impossible to do within *T. gondii*.

7. Binding affinities using ITC – TG CPSF4 with m6A has a Kd at 5uM for the modified oligo and Arabidopsis CPSF4 shows close to a Kd 6uM for the non-m6A modified oligo. Could the authors explain this judgement on what constitutes good/bad binding?

We have modified the ITC figures (these are the same titrations) to make the message less confusing (see new Figure 4). Indeed, Arabidopsis CPSF30-YTH titrations against non-m6A modified RNA were displayed without the same scale and we fitted a titration curve on a very weak shift in ΔH, in reality we can consider that there is no binding when we compare both.

Now for what is considered good or bad binding, what can be said is that in both cases specific binding only occurs with m6A.

8. In addition, the specificity of YTH binding to m6A is based on a single oligonucleotide sequence. At least a scrambled version of this should be used.

Aside from the technical difficulty to synthesize a scramble m6A modified oligo in quantities sufficient for ITC we don’t really see how this would benefit the central message of the ITC experiment, the main message being the centerstage importance of having m6A on a central position to measure any binding at all. This is also the same RNA which is seen within the crystal. Measuring a scrambled un-modified sequence against a scrambled modified sequence would probably not improve the affinity as the optimal sequences would not be in saturation quantities and eventually you would start titrating non optimal sequences as well, we would also lose the information related to the positioning of the m6A.

9. Based on the crystallization results where CPSF4 does not show RNA context specificity, except for m6A residue, how do the authors predict that this recruitment IS context specific? There exist other m6A sites in regions that do not constitute the 3' UTR (Line 612).

Context is probably controlled by PAS binding ability of the N-terminal zinc fingers of CPSF4, RNA secondary structures which may inhibit proper m6A recognition or a number of additional regulators such as CSF factors for which we identified a number of putative subunits within our MS purification of CPSF1.

10. Figure 7A -An IFA is not sufficient. A western blot is needed in the very least to validate the KD.

As suggested, we added a time-course western blot analysis of the expression levels of CPSF30–mAID–HA, using relevant loading controls (new Figure 7B).

11. Figure 7B – Are these adjusted p-values. Then these would not be a relatively low number of differentially expressed genes then as per Line 457.

We agree that since these are adjusted p-values the mention "relatively small number" is not relevant and we have now removed it from the main text.

12. Line 480 – There must be a way to quantify these readthrough events. A statement like this requires substantiation in the form of data.

As discussed previously, we have revisited the data to more clearly quantity chimeric reads.

13. Please provide a quantitative methodology to depict how METTL3 KD phenocopies CPSF4 KD. How many transcripts display aberrant splicing and which of these are common to both KDs?

We have answered this point which was raised above and have corrected the Figure 8 to include common and exclusive readthrough events in all the different mutants.

14. Line 209 – Could you draw a venn diagram or similar representation to see the overlapping proteins identified in the complex in each pulldown?

A Venn diagram (new Figure 1B) now details overlapping proteins identified in each pull-down.

15. Since you identified Val522 as an additional amino acid for the binding cage of m6A when compared to its counterparts, I suggest you to test the binding affinity towards m6A-modified RNA using V522 mutation by ITC.

We thank the reviewer for the suggestion but at this stage we do not believe that demonstrating the role played by the V522 is of paramount importance for the core message of the paper.